# Beta human papillomavirus 8E6 promotes alternative end joining

Changkun Hu[1,2], Taylor Bugbee[2], Rachel Palinski[3], Ibukun A Akinyemi[4,5], Michael T McIntosh[4], Thomas MacCarthy[6], Sumita Bhaduri-McIntosh[4,5], Nicholas Wallace[2]*

[1]Basic Sciences Division, Fred Hutchinson Cancer Research Center, Seattle, United States; [2]Division of Biology, Kansas State University, Manhattan, United States; [3]Veterinary Diagnostic Laboratory, Kansas State University, Manhattan, United States; [4]Child Health Research Institute, Department of Pediatrics, University of Florida, Gainesville, United States; [5]Department of Molecular Genetics and Microbiology, University of Florida, Gainesville, United States; [6]Laufer Center for Physical and Quantitative Biology, Stony Brook University, Stony Brook, United States

**Abstract** Double strand breaks (DSBs) are one of the most lethal DNA lesions in cells. The E6 protein of beta-human papillomavirus (HPV8 E6) impairs two critical DSB repair pathways: homologous recombination (HR) and non-homologous end joining (NHEJ). However, HPV8 E6 only delays DSB repair. How DSBs are repaired in cells with HPV8 E6 remains to be studied. We hypothesize that HPV8 E6 promotes a less commonly used DSB repair pathway, alternative end joining (Alt-EJ). Using CAS9-based Alt-EJ reporters, we show that HPV8 E6 promotes Alt-EJ. Further, using small molecule inhibitors, CRISPR/CAS9 gene knockout, and HPV8 E6 mutant, we find that HPV8 E6 promotes Alt-EJ by binding p300, an acetyltransferase that facilitates DSB repair by HR and NHEJ. At least some of this repair occurs through a subset of Alt-EJ known as polymerase theta dependent end joining. Finally, whole genome sequencing analysis showed HPV8 E6 caused an increased frequency of deletions bearing the microhomology signatures of Alt-EJ. This study fills the knowledge gap of how DSB is repaired in cells with HPV8 E6 and the mutagenic consequences of HPV8 E6 mediated p300 destabilization. Broadly, this study supports the hypothesis that beta-HPV promotes cancer formation by increasing genomic instability.

*For correspondence:
nwallac@ksu.edu

Competing interest: The authors declare that no competing interests exist.

## Editor's evaluation

This article reports useful data on how human papillomavirus 8E6 protein regulates DSB repair pathways in human cells. The data support the claim that 8E6 promotes alternative end-joining through binding and destabilizing the p300 acetyltransferase, showing the involvement of PARP-1-dependent alternative end-joining and to a lesser degree DNA Polymerase theta-dependent alternative end-joining.

## Introduction

Beta genus human papillomaviruses (beta-HPVs) are ubiquitous and transiently infect cutaneous epithelia in the general population (*Bouwes Bavinck et al., 2018*; *Gheit, 2019*; *Neale et al., 2013*). Beta-HPVs, including type 8 (HPV8), are associated with nonmelanoma skin cancer (NMSC) in immunocompromised individuals including people with a rare genetic disorder epidermodysplasia verruciformis and organ transplant recipients (*Bouwes Bavinck et al., 2018*; *Bouwes Bavinck et al., 2001*; *Dell'Oste et al., 2009*; *Sichero et al., 2019*). However, the contribution of beta-HPV infections to

**Figure 1.** 8E6 promotes alternative end joining (Alt-EJ) frequency. (**A**) Schematic of Alt-EJ reporter. GFP is disrupted by a 46 nt insertion. One CAS9 is used to induce an upstream double strand break (DSB) (5' end) and another CAS9 is used to induce a downstream DSB (either imbedded or terminal). Following CAS9 expression, a 4 nt microhomology (ACGG) mediated Alt-EJ event can restore GFP expression. (**B**) Representative images of flow cytometry results of HFK cells that are GFP positive 24 hr after transfection with terminal Alt-EJ. The gating represents GFP positive based off mock transfected control. The x-axis shows cells distributed by forward scatter to avoid debris. (**C**) Percentage of HFK cells that are positive for GFP following transfection with terminal Alt-EJ determined by flow cytometry. (**D**) Representative images of flow cytometry results of HFK cells that are GFP positive

*Figure 1 continued on next page*

*Figure 1 continued*

24 hr after transfection with imbedded Alt-EJ. The gating represents GFP positive based off mock transfected control. The x-axis shows cells distributed by forward scatter to avoid debris. (**E**) Percentage of HFK cells that are positive for GFP following transfection with imbedded Alt-EJ determined by flow cytometry. All values are represented as mean ± standard error. The statistical significance of differences between cell lines were determined using Student's t-test. p-Values indicate significant difference between cell lines. Twenty thousand cells were counted for each of three independent flow cytometry experiments.

The online version of this article includes the following source data and figure supplement(s) for figure 1:

**Figure supplement 1.** Transfection efficiency represented by CAS9 expression in hTERT human foreskin keratinocyte (HFK).

**Figure supplement 1—source data 1.** Original blots with and without labels in *Figure 1—figure supplement 1*.

NMSC in the general population is unclear. Because beta-HPV infections are not persistent in immuno-competent people, they are hypothesized to promote cancer formation by making UV-induced DNA damage more mutagenic (*Hufbauer et al., 2015*; *Spriggs and Laimins, 2017*; *Wendel and Wallace, 2017*). In support of this hypothesis, our group and others have used in vitro and in vivo systems to demonstrate that the E6 protein from HPV8 (8E6) impairs DNA repair (*Hufbauer et al., 2015*; *Rollison et al., 2019*; *Wallace et al., 2012*).

The interaction with and destabilization of p300 is a key mechanism by which 8E6 hinders DNA repair (*Dacus and Wallace, 2021*; *Howie et al., 2011*). P300 is an acetyltransferase that regulates transcription by chromatin remodeling (*Ait-Si-Ali et al., 2000*; *Dutto et al., 2018*; *Goodman and Smolik, 2000*). By binding p300, 8E6 decreases the abundance of multiple DNA repair proteins including ATR, ATM, BRCA1, and BRCA2 (*Wallace et al., 2015*; *Wallace et al., 2013*; *Wallace et al., 2012*). This lowers the activation of ATM and ATR signaling, decreasing the cellular response to UV-damaged DNA (*Snow et al., 2019*; *Wallace et al., 2012*). The limited ability to repair UV damage increases the frequency with which UV causes replication forks to collapse into double strand breaks (DSBs) in DNA (*Pathania et al., 2011*; *Rapp and Greulich, 2004*). Cells have multiple DSB repair mechanisms. Homologous recombination (HR) is minimally mutagenic, but restricted to in S/G2 phase when the sister chromatids can serve as homologous templates (*Orthwein et al., 2015*; *Shibata et al., 2011*). Whenever possible cells use HR to fix DSBs as it allows them to avoid mutations. When HR is inhibited (by cell cycle position, mutation to repair factors, or artificially), non-homologous end joining (NHEJ) is used to repair DSBs (*Wang et al., 2018*). NHEJ can occur throughout the cell cycle, as it does not require a homologous template. However, because NHEJ generates and ligates blunt ends to fix a DSB, it is more mutagenic than HR (*Mao et al., 2008*). We have shown that 8E6 attenuates both HR and NHEJ by degrading p300 (*Hu et al., 2020*; *Hu and Wallace, 2022c*; *Wallace et al., 2015*).

Notably, 8E6 does not block the initiation of NHEJ or HR. NHEJ initiation occurs when DNA-dependent protein kinase catalytic subunit (DNA-PKcs) complexes at DSBs and activates itself by autophosphorylation (pDNA-PKcs) (*Davis et al., 2014*; *Jiang et al., 2015*). This occurs readily in the presence of 8E6. However, 8E6 prevents the resolution of pDNA-PKcs repair complexes and attenuates other downstream steps in NHEJ (*Hu et al., 2020*). Similarly, 8E6 allows the HR pathway to initiate, before hindering the resolution of RAD51 repair complexes (*Wallace et al., 2015*). We recently demonstrated that cells respond to 8E6-associated inhibition of NHEJ by trying to complete HR during G1 (*Hu et al., 2022a*; *Hu et al., 2022b*). This ultimately leads to persistent unresolved RAD51 repair complexes formed during G1.

Thus, currently there is a detailed understanding of how 8E6 causes DSB repair to fail, but less is known about how DSB repair occurs in cells that express 8E6. When NHEJ and HR fail, another mutagenic repair pathway known as alternative end joining (Alt-EJ) is tasked with completing DSB repair (*Iliakis et al., 2015*). Here, we use reporter constructs and small molecule inhibitors of DNA repair factors to demonstrate that 8E6 promotes DSB repair by Alt-EJ and to show that the use of Alt-EJ is the indirect result of initiating NHEJ when the pathway cannot be completed. We also employ whole genome sequence analysis of cells expressing 8E6 and passage matched empty vector control expressing cells to determine the frequency with which 8E6 promotes mutations with the characteristics of repair by Alt-EJ. These observations address a key knowledge gap in the field. By promoting DSB repair by Alt-EJ, 8E6 increases the risk of mutations associated with DSBs while allowing cells to avoid the apoptosis that would be associated with an unrepaired DSB. This is consistent with the proposed mechanism by which beta-HPV infections are hypothesized to promote NMSC.

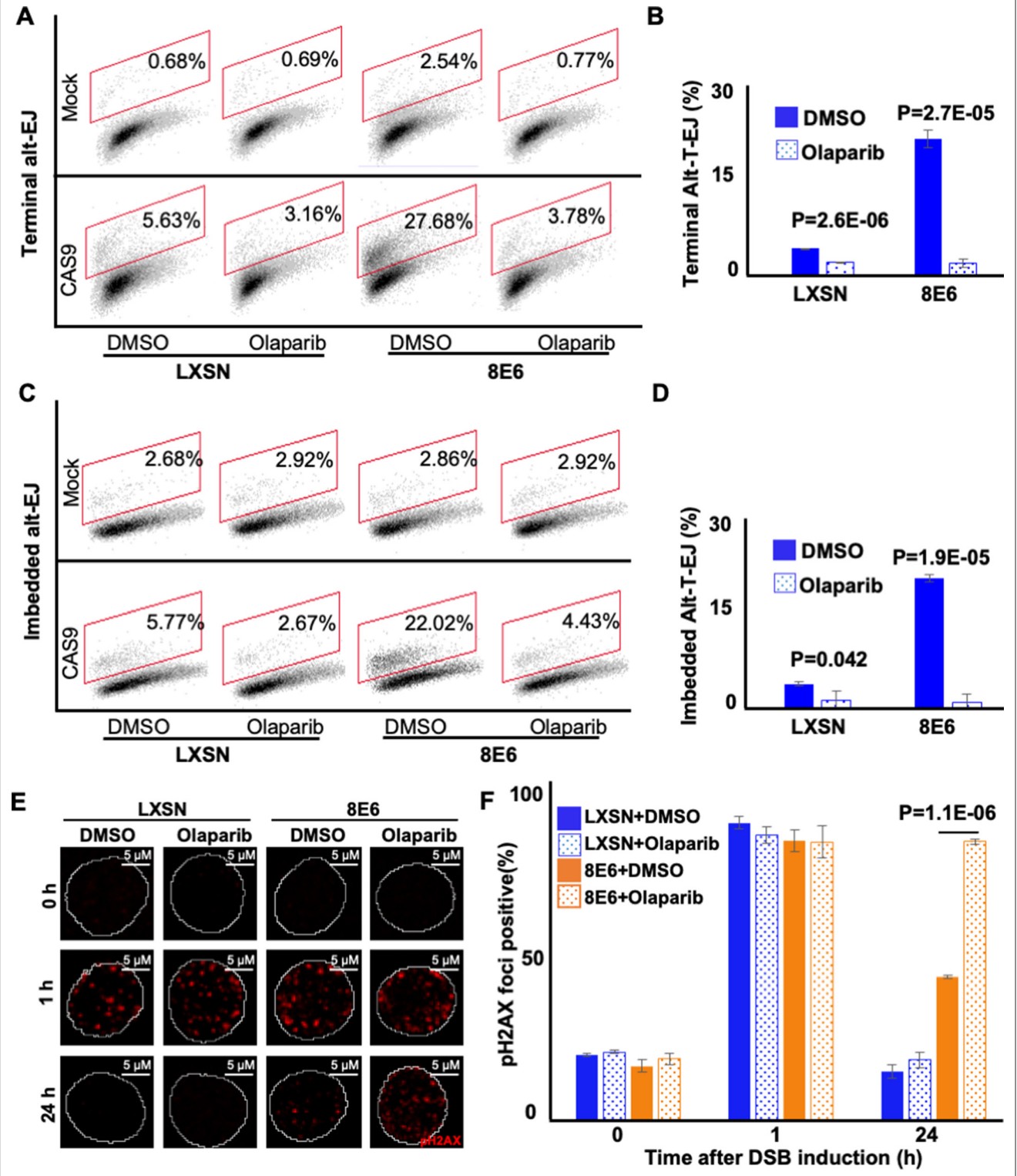

**Figure 2.** Olaparib abrogates alternative end joining (Alt-EJ) frequency and increases persistent pH2AX. (**A**) Representative images of flow cytometry results of human foreskin keratinocyte (HFK) cells treated with DMSO or olaparib (1 μM) that are GFP positive 24 hr after transfection with terminal Alt-EJ. The gating represents GFP positive based off mock transfected control. The x-axis shows cells distributed by forward scatter to avoid debris. (**B**) Percentage of HFK cells that are positive for GFP following transfection with terminal Alt-EJ determined by flow cytometry. (**C**) Representative images of flow cytometry results of HFK cells treated with DMSO or olaparib that are GFP positive 24 hr after transfection with imbedded Alt-EJ. The gating represents GFP positive based off mock transfected control. The x-axis shows cells distributed by forward scatter to avoid debris. (**D**) Percentage of HFK

*Figure 2 continued on next page*

*Figure 2 continued*

cells that are positive for GFP following transfection with imbedded Alt-EJ determined by flow cytometry. (**E**) Representative images of pH2AX in HFK LXSN and HFK 8E6 treated with DMSO or olaparib (1 µM) following zeocin treatment (10 µg/mL, 10 min). (**F**) Percentage of pH2AX foci positive cells in HFK LXSN and HFK 8E6 treated with DMSO or olaparib following zeocin treatment. All values are represented as mean ± standard error. The statistical significance of differences between treatments were determined using Student's t-test. p-Values indicate significant difference between DMSO and olaparib with same cell line (p<0.05). Twenty thousand cells were counted for each of three independent flow cytometry experiments.

The online version of this article includes the following source data and figure supplement(s) for figure 2:

**Figure supplement 1.** Transfection efficiency represented by CAS9 expression in hTERT human foreskin keratinocyte (HFK).

**Figure supplement 1—source data 1.** Original blots with and without labels in *Figure 2—figure supplement 1*.

## Results
### 8E6 promotes DSB repair by Alt-EJ

8E6 delays rather than abrogates DSB repair, but it is unclear how these lesions are repaired as 8E6 hinders the completion of HR and NHEJ. Because Alt-EJ serves as another DSB repair mechanism should HR and NHEJ fail, we hypothesized that DSBs were instead repaired by Alt-EJ. To test this, we examined previously described telomerase (N/TERT) immortalized human foreskin keratinocyte (HFK) expressing vector control (HFK LXSN) and 8E6 (HFK 8E6) (*Bedard et al., 2008*). An established reporter cassette where a 46 nt insertion disrupts a GFP open reading frame was used to measure Alt-EJ (*Bhargava et al., 2018*; *Tsai et al., 2020*). *Figure 1A* describes Alt-EJ that requires resection (imbedded) and Alt-EJ that occurs independent of resection (terminal). The end joining is mediated by a 4 nt microhomology (ACGG). Transient transfection of this reporter into HFK LXSN and HFK 8E6 demonstrated that 8E6 increased terminal and imbedded Alt-EJ (*Figure 1B–E* and *Figure 1—figure supplement 1*). Transfection efficiency varies between independent experiments, but no significant difference was observed among groups within each experiment (*Figure 1—figure supplement 1*).

To further confirm that 8E6 promoted Alt-EJ, we examined DSB repair in cells where Alt-EJ was blocked by a small molecule inhibitor against PARP-1, an established component of the Alt-EJ pathway (1 µM of olaparib) (*Kötter et al., 2014*). We confirmed that PARP-1 inhibition blocked Alt-EJ using the reporter system described above (*Figure 2A–D* and *Figure 2—figure supplement 1*). DSBs were induced by growth in media containing zeocin (10 µg/mL, 10 min), a radiation mimetic reagent (*Tsukuda and Miyazaki, 2013*). H2AX phosphorylated at Serine 139 or pH2AX is used as a standard DSB marker (*Rogakou et al., 1998*). PARP-1 inhibition did not significantly alter DSB repair in HFK LXSN cells (*Figure 2E–F*). This is consistent with the established view that most DSB repair occurs via either HR or NHEJ, with Alt-EJ serving as a fail-safe machinery should these pathways fail (*Iliakis et al., 2015*; *Patterson-Fortin and D'Andrea, 2020*; *Simsek and Jasin, 2010*). In contrast, PARP-1 inhibition significantly delayed DSB repair in HFK 8E6 cells (*Figure 2E–F*).

Alt-EJ is an umbrella term that includes any end joining event that does not require DNA-PKcs activity. While most Alt-EJ relies on PARP-1, a subset of Alt-EJ can occur when PARP-1 activity is impaired. Specifically, loss of PARP-1 activity only reduces polymerase theta mediated end joining (TMEJ) by two- to fourfold (*Luedeman et al., 2022*). To begin determining the extent that 8E6 promoted TMEJ, we used immunoblots to determine if 8E6 increased the abundance of polymerase theta (POLθ). This analysis did not find any notable changes in POLθ abundance (*Figure 3A–B*). We next used a small molecule inhibitor of POLθ (ART558) to determine if HFK 8E6 cells were reliant on POLθ activity, using an MTT assay to detect viability. There were no significant differences in viability between HFK LXSN and HFK 8E6 cells when grown in media containing a gradient (0–50 µM) of ART558 concentrations (*Figure 3C*) However, 8E6 impairs apoptosis by degrading BAK, which complicates the interpretation of these data. To more directly define the extent that 8E6 promotes DSB repair by TMEJ, we determined the extent that POLθ inhibition impaired DSB repair. For this analysis, we induced DSBs by growth media containing zeocin (10 µg/mL, 10 min). Cells were then switched to media containing DMSO (solvent control), 1 µM 558, or 5 µM ART 558. DSBs were detected by immunofluorescence microscopy of pH2AX24 hr later. These concentrations of ART558 were analyzed because they caused the largest differences in viability between HFK 8E6 and HFK LXSN cells. ART558 delayed DSB repair in HFK 8E6 cells but not in HFK LXSN cells, indicating that 8E6 promoted repair by TMEJ (*Figure 3D–E*). However, although statistically significant, the increased persistence of pH2AX foci was a much lower magnitude than what PARP1 inhibition caused. These experiments were also

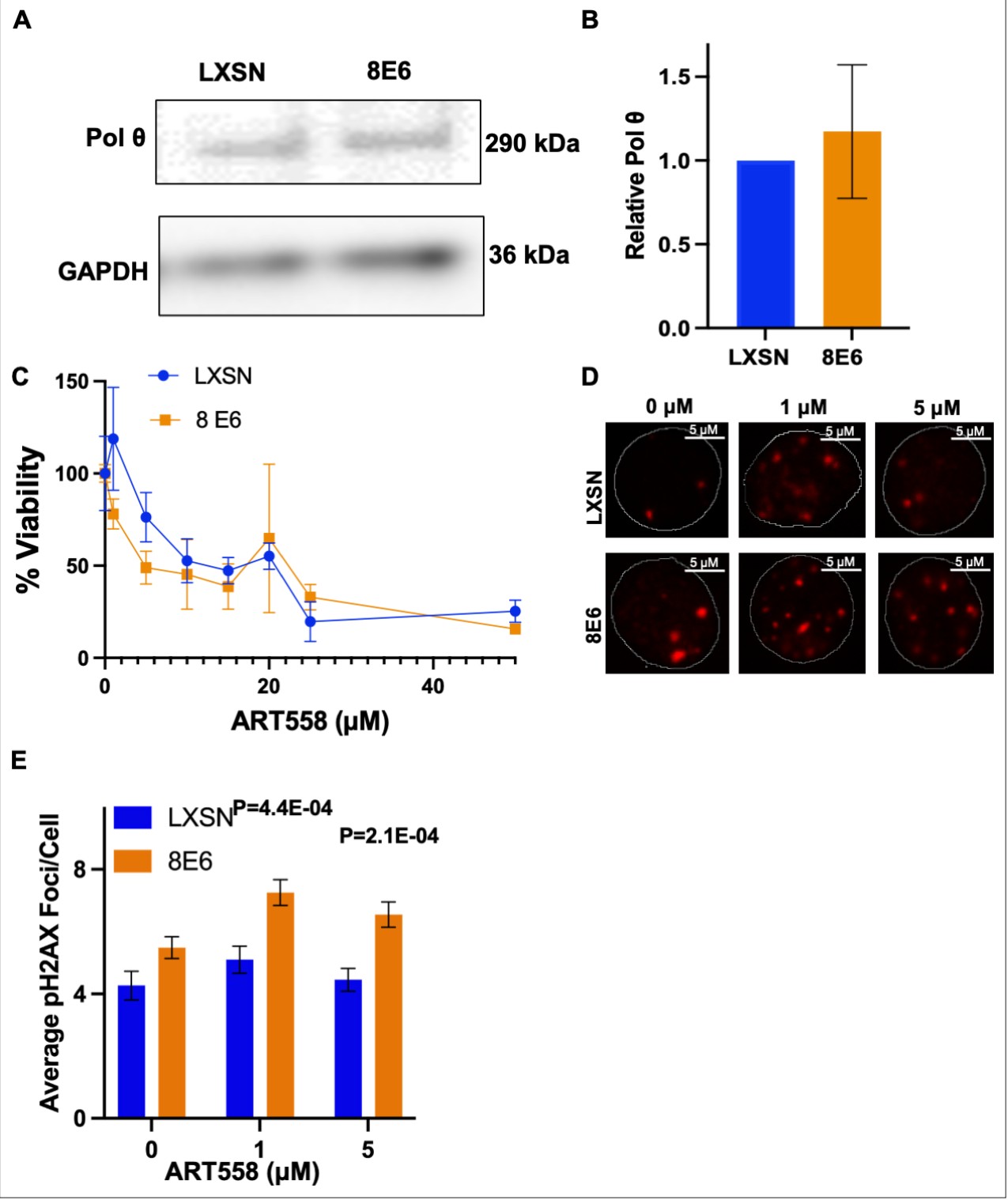

**Figure 3.** 8E6 promotes polymerase theta (POL $\theta$)-dependent double strand break (DSB) repair. (**A**) Representative immunoblotting of POL $\theta$ in human foreskin keratinocyte (HFK) LXSN and HFK 8E6 cells. (**B**) Densitometry of POL $\theta$ level in HFK LXSN and HFK 8E6 cells. (**C**) Relative cell viability at various ART558 concentrations in HFK LXSN and HFK 8E6 following zeocin treatment. (**D**) Representative images of pH2AX in HFK LXSN and HFK 8E6 treated with DMSO or ART588 (1 or 5 μM) 24 hr following zeocin treatment (10 μg/mL, 10 min). (**E**) Average number of pH2AX foci per cell in HFK LXSN and HFK 8E6 treated with DMSO or ART558 following zeocin treatment. The statistical significance of differences between treatments were determined using Student's t-test. p-Values indicate significant difference between HFK LXSN and HFK 8E6 with same ART588 treatment (p<0.05). At least 40 cells were counted for each of three independent microscopy experiments.

*Figure 3 continued on next page*

*Figure 3 continued*

The online version of this article includes the following source data for figure 3:

**Source data 1.** Original blots with and without labels in *Figure 3*.

repeated with higher concentrations of ART558, but no differences in the persistence of pH2AX was detected (not shown). We interpret these data as evidence that 8E6 inhibits multiple types of DNA-PKcs-independent end joining, including but not limited to TMEJ. For simplicity, we will only use the term Alt-EJ moving forward.

## DNA-PK inhibition forces 8E6 expressing cells to use Alt-EJ more frequently

NHEJ initiation blocks DSB repair by other pathways, including Alt-EJ (*Patterson-Fortin and D'Andrea, 2020*; *Simsek and Jasin, 2010*; *Wang et al., 2006*). 8E6 does not prevent the initiation of NHEJ as autophosphorylated DNA-PKcs foci form readily in HFK 8E6 cells (*Hu et al., 2020*). Instead, 8E6 blocks the completion of NHEJ. Thus, we hypothesized that preventing HFK 8E6 cells from initiating NHEJ would force them to repair DSBs via Alt-EJ. To test this, we determined the frequency of Alt-EJ in HFK LXSN and HFK 8E6 cells in the presence of a small molecule inhibitor of DNA-PKcs (1 µM of NU7441) to block NHEJ initiation. As expected, DNA-PKcs inhibition increased both imbedded and terminal Alt-EJ in HFK LXSN cells (*Figure 4* and *Figure 4—figure supplement 1*). However, the increase in HFK LXSN cells did not reach the levels of Alt-EJ in mock-treated HFK 8E6 cells. DNA-PKcs inhibition resulted in a further significant increased use of Alt-EJ in HFK 8E6 cells. Together these data support the conclusion that 8E6 promotes the use of Alt-EJ and that the use of the pathway is further enhanced if cells are not allowed to initiate NHEJ.

When HFK 8E6 cells are allowed to initiate NHEJ, many of the pDNA-PKcs repair complexes that form persist for over 24 hr (*Hu et al., 2020*). HR factors (e.g., RAD51) are then recruited to these unresolved repair foci (*Hu et al., 2022a*). However, HR is ultimately unable to repair the lesions. We hypothesized that forcing HFK 8E6 cells to use Alt-EJ by preventing initiation of NHEJ would lead to more efficient DSB repair by allowing them to avoid these abortive attempts at DSB repair. NU7441 (1 mM) was used to block NHEJ initiation and DSBs were detected using pH2AX. As we have shown before, DSBs were more persistent in mock-treated 8E6 cells than LXSN cells. Supporting our hypothesis, DNA-PKcs inhibition made DSB repair more efficient (less pH2AX) in HFK 8E6 cells (*Figure 5A–B*).

We next determined the extent that DNA-PKcs inhibition in HFK 8E6 cells resulted in increased resolution of RAD51 foci. As we have previously reported (*Hu et al., 2022a*), DNA-PKcs inhibition increased the persistence of RAD51 in HFK LXSN (*Figure 5C–D*). A similar increase in RAD51 persistence was also seen in mock-treated HFK 8E6 cells. However, DNA-PKcs inhibition increased the rate of RAD51 resolution in HFK 8E6 (*Figure 5C–D*). To determine if inhibition of a later step in the NHEJ pathway also increased the efficiency of DSB repair in HFK 8E6 cells, we used a small molecule inhibitor of ligase IV (1 µM of SCR7) to block a near terminal step in NHEJ. Ligase IV inhibition delayed DSB repair in both HFK LXSN and HFK 8E6 cells (*Figure 5E–F*). Thus, restoration of DSB repair in HFK 8E6 specifically requires inhibition of an early NHEJ step, rather than inhibition of a later step in the pathway.

## DNA-PKcs inhibition prevents 8E6 from causing RAD51 foci to form during G1

The persistent RAD51 repair complexes that form in HFK 8E6 cells occur during G1 (*Hu et al., 2022a*). These observations and the ones described in *Figure 4* led us to hypothesize that blocking NHEJ initiation via DNA-PKcs inhibition would prevent 8E6 from causing RAD51 foci to form in G1. To test this, we detected RAD51 foci and cyclin E (G1 marker) after the induction of DSB by zeocin. DNA-PKcs inhibition increased the frequency of HFK LXSN in G1 that contained RAD51 foci (*Figure 6A–B*). Without DNA-PKcs inhibition, 8E6 increased the frequency of cells in G1 that had RAD51 foci. However, DNA-PKcs inhibition prevented 8E6 from promoting the formation of RAD51 repair complexes during G1. As cyclin E is also expressed during early S phase, this examination was repeated using cyclin A as a marker of cells in S/G2 (*Figure 6—figure supplement 1*). These experiments confirmed our observations using cyclin E to determine cell cycle position, providing further evidence that DNA-PKcs

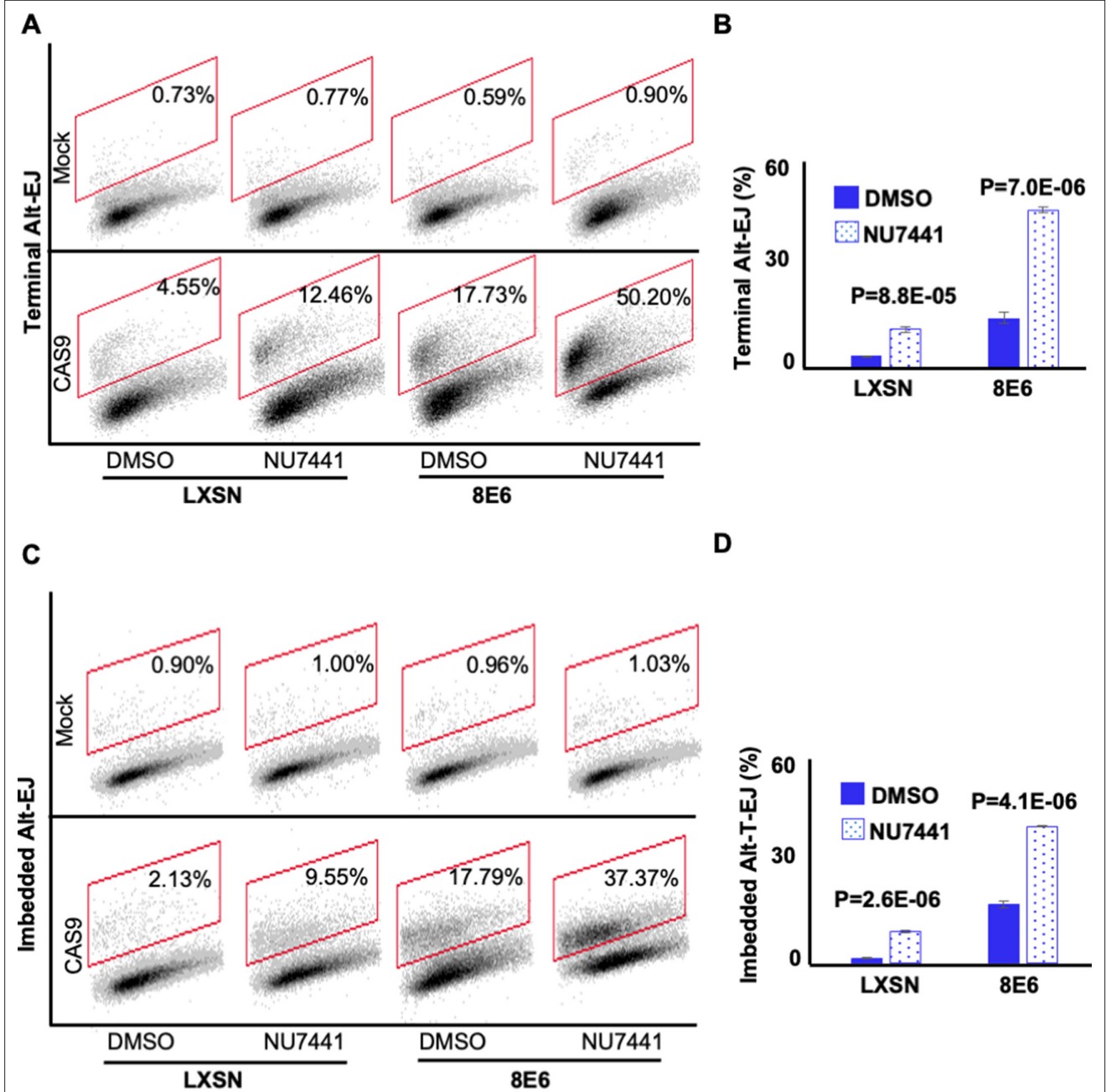

**Figure 4.** NU7441 promotes alternative end joining (Alt-EJ). (**A**) Representative images of flow cytometry results of human foreskin keratinocyte (HFK) cells treated with DMSO or NU7441 (1 µM) that are GFP positive 24 hr after transfection with terminal Alt-EJ. The gating represents GFP positive based off mock transfected control. The x-axis shows cells distributed by forward scatter to avoid debris. (**B**) Percentage of HFK cells that are positive for GFP following transfection with terminal Alt-EJ determined by flow cytometry. (**C**) Representative images of flow cytometry results of HFK cells treated with DMSO or NU7441 that are GFP positive 24 hr after transfection with imbedded Alt-EJ. The gating represents GFP positive based off mock transfected control. The x-axis shows cells distributed by forward scatter to avoid debris. (**D**) Percentage of HFK cells that are positive for GFP following transfection with imbedded Alt-EJ determined by flow cytometry. All values are represented as mean ± standard error. The statistical significance of differences between treatments were determined using Student's t-test. p-Values indicate significant difference between DMSO and NU7441 within the same cell line. Twenty thousand cells were counted for each of three independent flow cytometry experiments.

The online version of this article includes the following source data and figure supplement(s) for figure 4:

**Figure supplement 1.** Transfection efficiency represented by CAS9 expression in hTERT human foreskin keratinocyte (HFK).

**Figure supplement 1—source data 1.** Original blots with and without labels in *Figure 4—figure supplement 1*.

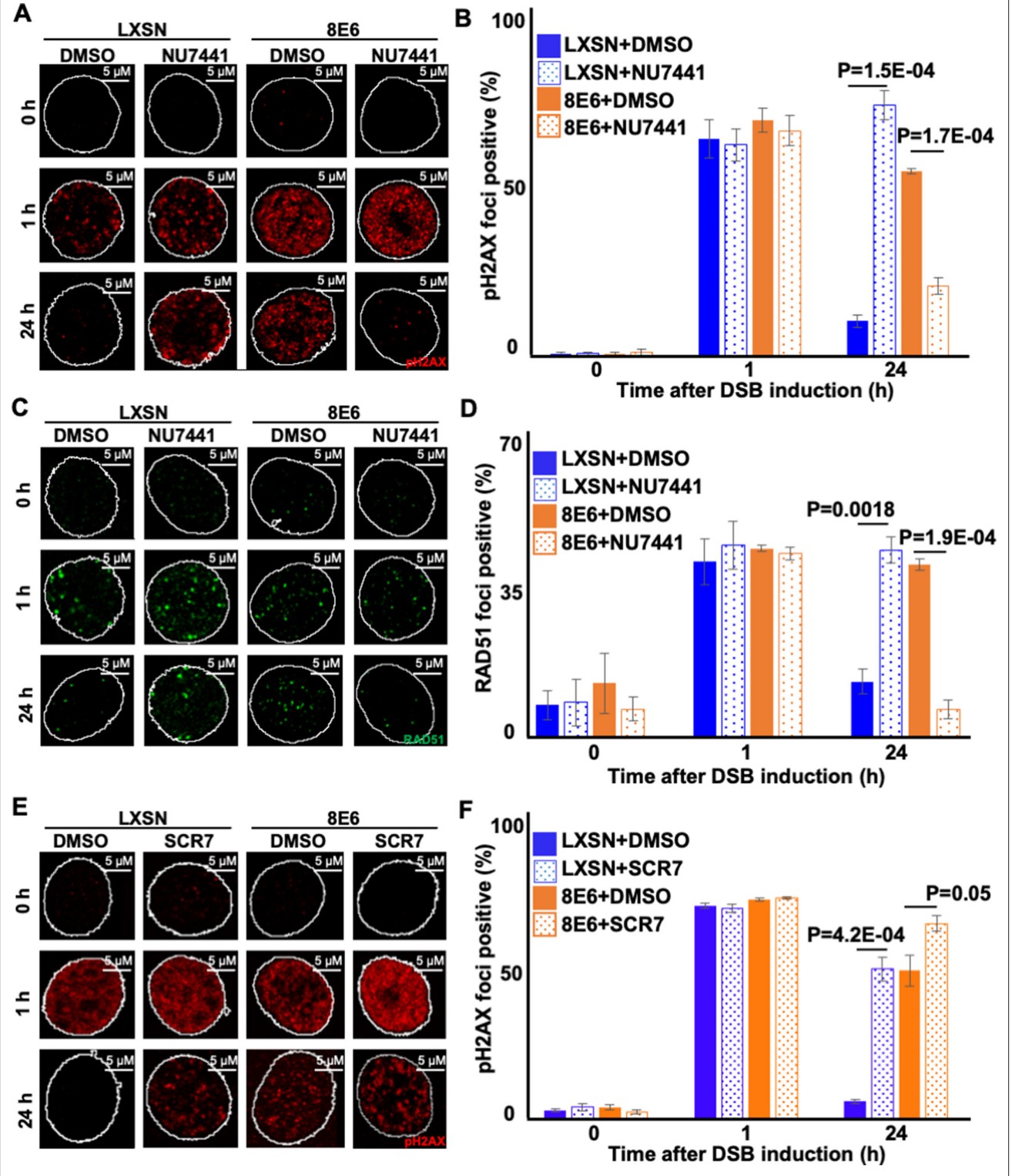

**Figure 5.** Nu7441 increases double strand break (DSB) repair in cells with 8E6. (**A**) Representative images of pH2AX in human foreskin keratinocyte (HFK) LXSN and HFK 8E6 treated with NU7441 (1 µM) following zeocin treatment (10 µg/mL, 10 min). (**B**) Percentage of pH2AX foci positive cells in HFK LXSN and HFK 8E6 treated with NU7441 following zeocin treatment. (**C**) Representative images of RAD51 in HFK LXSN and HFK 8E6 treated with NU7441 following zeocin treatment. (**D**) Percentage of RAD51 foci positive cells in HFK LXSN and HFK 8E6 treated with NU7441 following zeocin treatment. (**E**) Representative images of pH2AX in HFK LXSN and HFK 8E6 treated with SCR7 (1 µM) following zeocin treatment. (**F**) Percentage of pH2AX foci positive cells in HFK LXSN and HFK 8E6 treated with SCR7 following zeocin treatment. All values are represented as mean ± standard error.

*Figure 5 continued on next page*

*Figure 5 continued*

The statistical significance of differences between treatments were determined using Student's t-test. p-Values indicate significant difference between DMSO and inhibitor treated with the same cell line. At least 150 cells were counted over three independent experiments. Nuclei were determined by DAPI staining. The edge of this staining is shown by a white line depicting the nucleus.

inhibition prevented 8E6 from promoting the formation of RAD51 foci during G1 (*Figure 6—figure supplement 1*). We used flow cytometry as a final determinant of cell cycle position, using NUCLE-AR-ID Red DNA staining to select cells in G1 based on DNA content and then determined the frequency with which these cells stained for RAD51 (*Figure 6—figure supplement 2*). Consistent with our hypothesis, DNA-PKcs inhibition increased the frequency of RAD51 in G1 in HFK LXSN and prevented 8E6 from promoting RAD51 in G1 (*Figure 6C–D*).

## DNA-PKcs inhibition does not promote HR in cells with 8E6

The data above demonstrate that the attenuation of DSB repair by 8E6 can be overcome by inhibiting DNA-PKcs. DNA-PKcs inhibition also increases the resolution of RAD51, suggesting that DNA-PKcs inhibition may prevent 8E6 from attenuating HR. To test this, we measured HR efficiency using an established HR reporter, described in *Figure 7A* (*Pierce et al., 1999*). In vector control U2OS cells (U2OS LXSN), NU7441 increases HR efficiency (*Figure 7B–C*). This is consistent with the established idea that NHEJ and HR compete for access to DSBs (*Bunting et al., 2010*; *Chapman et al., 2012*; *Orthwein et al., 2015*). As previously reported, 8E6 in U2OS cells (U2OS 8E6) decreased HR efficiency (*Wallace et al., 2015*). However, DNA-PKcs inhibition did not prevent 8E6 from hindering HR (*Figure 7B–C*). Consistent with a p300-dependent mechanism (see next section for more details), U2OS cells expressing 8E6 with the residues responsible for binding p300 deleted (U2OS 8E6Δ132–136) were similar to vector control (*Figure 7B–C*). These cell lines have been previously described (*Hu et al., 2020*; *Wallace et al., 2015*). U2OS cells are routinely used to probe 8E6 biology because 8E6 retains its ability to alter DNA repair in these cells (*Hu et al., 2020*; *Wallace et al., 2015*).

## 8E6 promotes Alt-EJ by destabilizing p300

8E6 delays DSB repair by binding/destabilizing p300, leading us to hypothesize that the residues of 8E6 that facilitate the interaction with p300 were important for the phenotypes described thus far (*Hu et al., 2022a*, *Hu et al., 2020*; *Wallace et al., 2015*). To test this, we examined U2OS LXSN, U2OS 8E6, and U2OS 8E6Δ132–136 cells. Consistent with the p300-dependent mechanism, 8E6Δ132–136 shows similar Alt-EJ frequency with U2OS LXSN (*Figure 7A–B*). The deletion of these residues from 8E6 has been shown to prevent some but not all aspects of 8E6 biology. As a result, we examined p300 knockout N/TERT immortalized HFK cells. P300 knockout led to increases in both terminal and imbedded Alt-EJ (*Figure 8C–D*). To further confirm p300 dependence, we used a small molecule inhibitor of p300 (1 µM of CCS1477) to block p300 activity. Consistently, CCS1477 increased both terminal and imbedded Alt-EJ (*Figure 8E–F*).

## DNA-PKcs inhibition prevents the formation of RAD51 foci in G1 that is caused by loss of p300

We next confirmed that the ability of 8E6 to allow RAD51 foci to form in G1 was blocked by DNA-PKcs inhibition in U2OS cells (*Figure 9A–B*). Consistent with a p300-dependent mechanism, RAD51 foci were more likely to occur in G1 when DNA-PKcs was inhibited in U2OS LXSN and U2OS 8E6Δ132–136 cells. Further confirming a p300-dependent mechanism of action, HFK without p300 displayed an increased frequency of RAD51 staining in G1 that could be overcome by DNA-PKcs inhibition (*Figure 9C–D*). As a final confirmation of the p300 dependence of this phenotype, we treated N/TERT HFKs with 1 µM of CCS1477 to block p300 activity. Inhibition of p300 alone increased the frequency of RAD51 staining in G1 as did DNA-PKcs inhibition alone (*Figure 9E–F*). However, RAD51 staining in G1 was not increased by their dual application.

## Alt-EJ deletions occur at a higher frequency in 8E6 expressing cells

HFK cells were transfected with the 8E6 expression vector or empty vector and equally passaged under selective pressure to establish HFK 8E6 versus control HFK LXSN cell lines. These were

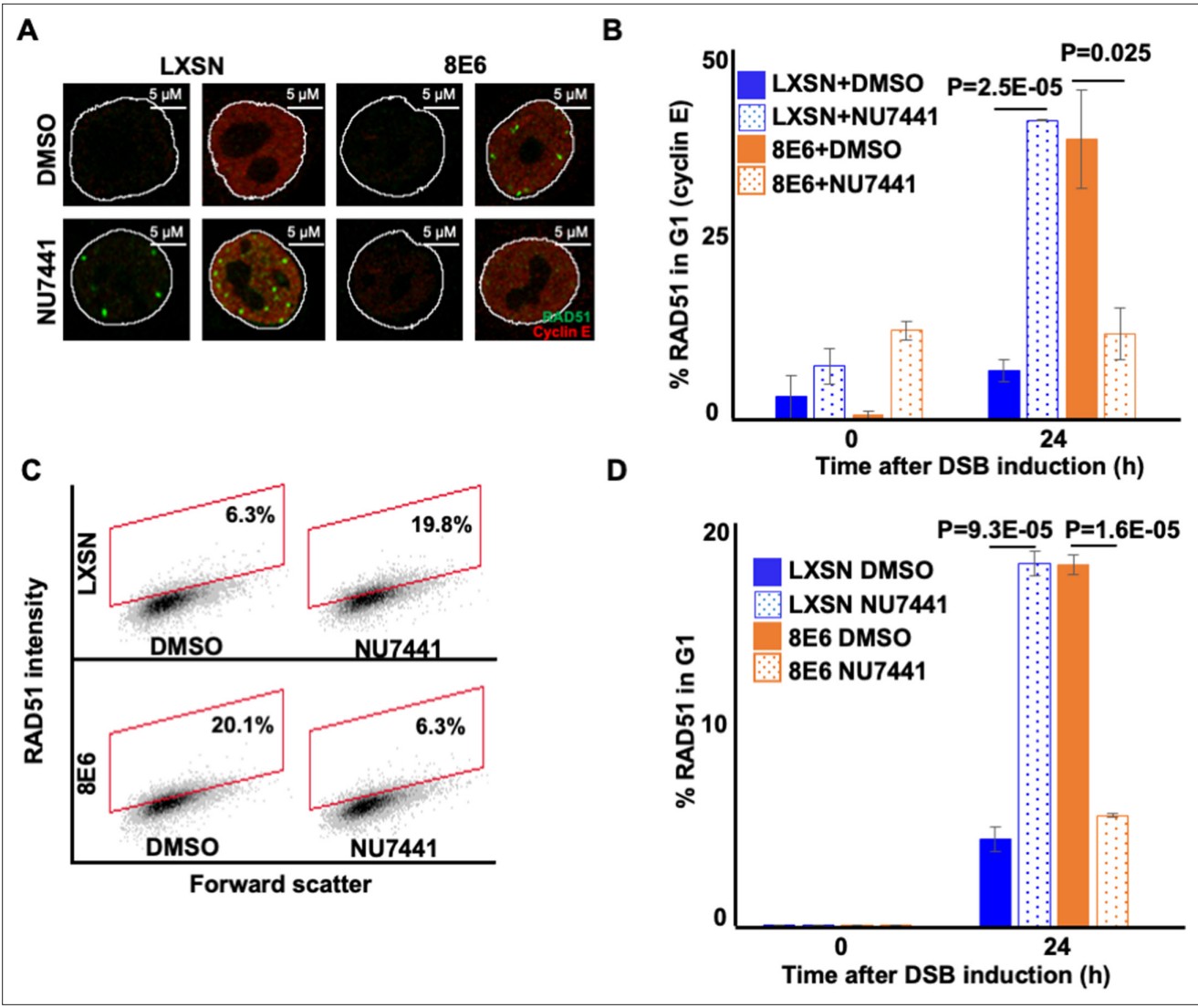

**Figure 6.** NU7441 abrogates RAD51 in G1 induced by 8E6. (**A**) Representative cyclin E negative and positive human foreskin keratinocyte (HFK) LXSN and HFK 8E6 cells stained for RAD51 (green) and cyclin E (red) treated with DMSO or NU7441 (1 µM) 24 hr following zeocin treatment (10 µg/mL, 10 min). (**B**) Percentage of RAD51 positive HFK cell in G1 determined by cyclin E staining after zeocin treatment. (**C**) Representative images of flow cytometry results of HFK LXSN and HFK 8E6 cells in G1 stained with RAD51 treated with DMSO or NU7441 24 hr after zeocin treatment. RAD51 intensity is determined by Alexa 488-conjugated secondary antibody and shown on the y-axis. The gating represents RAD51 positive based off secondary only control. The x-axis shows cells distributed by forward scatter to avoid debris. (**D**) Percentage of HFK cells in G1 that are positive for RAD51 as determined by flow cytometry. Nuclei were determined by DAPI staining. The edge of this staining is shown by a white line depicting the nucleus. All values are represented as mean ± standard error. The statistical significance of differences between treatments were determined using Student's t-test. p-Values indicate significant difference between DMSO and NU7441 treatment with the same cell line. At least 150 cells were counted over three independent experiments for microscopy. Twenty thousand cells were counted for each of three independent flow cytometry experiments.

The online version of this article includes the following figure supplement(s) for figure 6:

**Figure supplement 1.** NU7441 abrogates RAD51 in G1 (cyclin A negative) induced by 8E6.

**Figure supplement 2.** Controls were used to determine RAD51 staining cutoff and G1 gating in human foreskin keratinocyte (HFK) cells by flow cytometry.

subjected to whole genome sequencing to determine the type and frequency of mutations accumulated within each cell type. This yielded 933,163,972 high-quality paired-end sequence reads for LXSN and 624,472,714 reads for 8E6 (*Table 1*). Normalizing LXSN reads to the same number as 8E6 and alignment to the human genome yielded 92.4% genome coverage for both and slightly more read depth for LXSN (mean 45.4 for LXSN compared to 44.6 for 8E6) (*Table 2*). Variant calling including

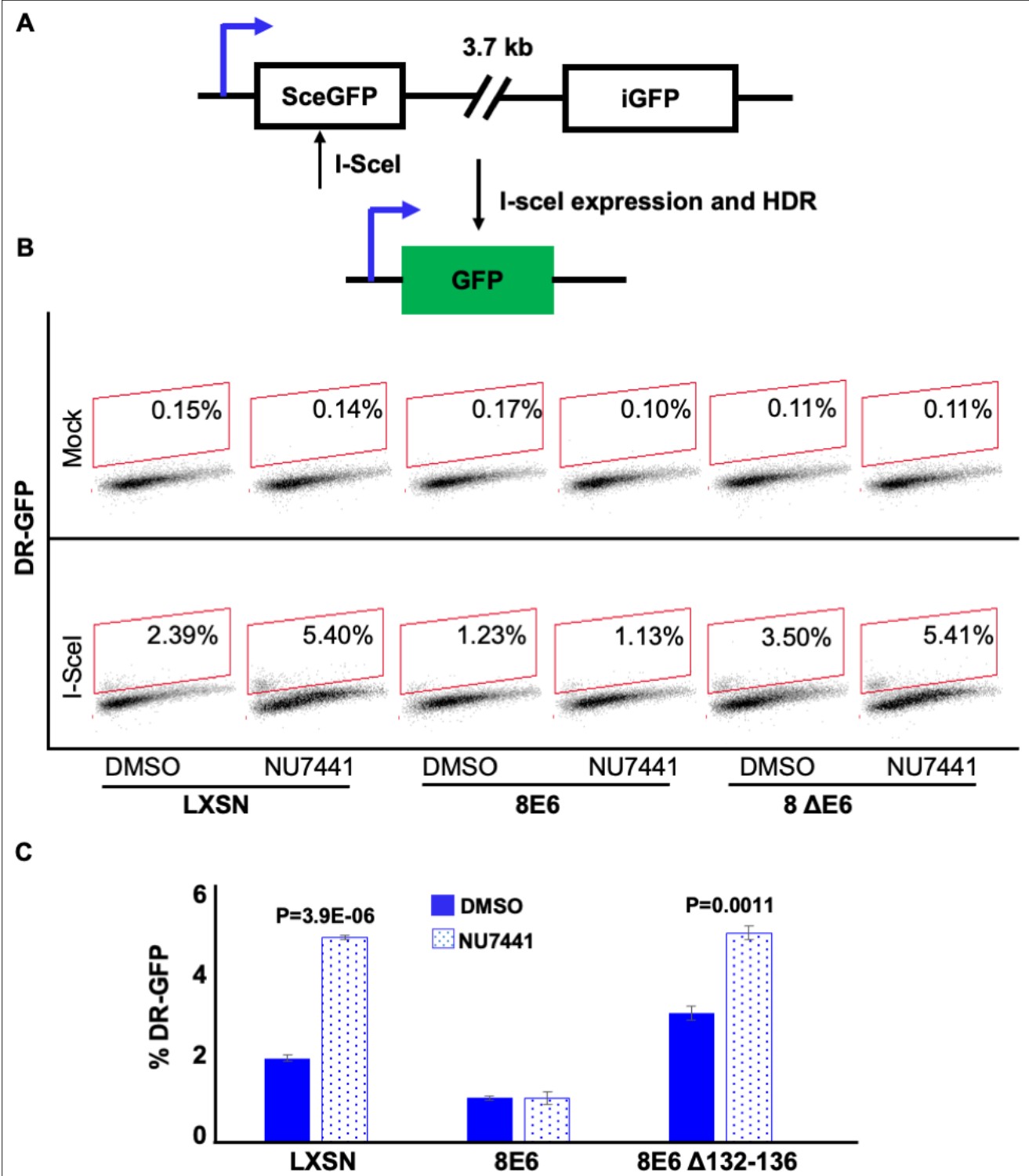

**Figure 7.** NU7441 does not increase homologous recombination (HR) in cells with 8E6. (**A**) Schematic of DR-GFP reporter. GFP open reading frame is disrupted by insertion of ISCE-1 site (SceGFP). Downstream of the reporter is a truncated internal GFP(iGFP) that can be used as a template to remove the ISCE-1 site and restore GFP expression during HR event. (**B**) Representative images of flow cytometry results of U2OS cells that are GFP positive treated with DMSO or NU7441(1 μM) 24 hr after ISCE-1 transfection. The gating represents GFP positive based off mock transfected control. The x-axis shows cells distributed by forward scatter to avoid debris. (**C**) Percentage of U2OS cells that are positive for GFP determined by flow cytometry. All values are represented as mean ± standard error. The statistical significance of differences between treatments were determined using Student's t-test. p-Values indicate significant difference between DMSO and NU7441 with same cell line. Twenty thousand cells were counted for each of three independent flow cytometry experiments.

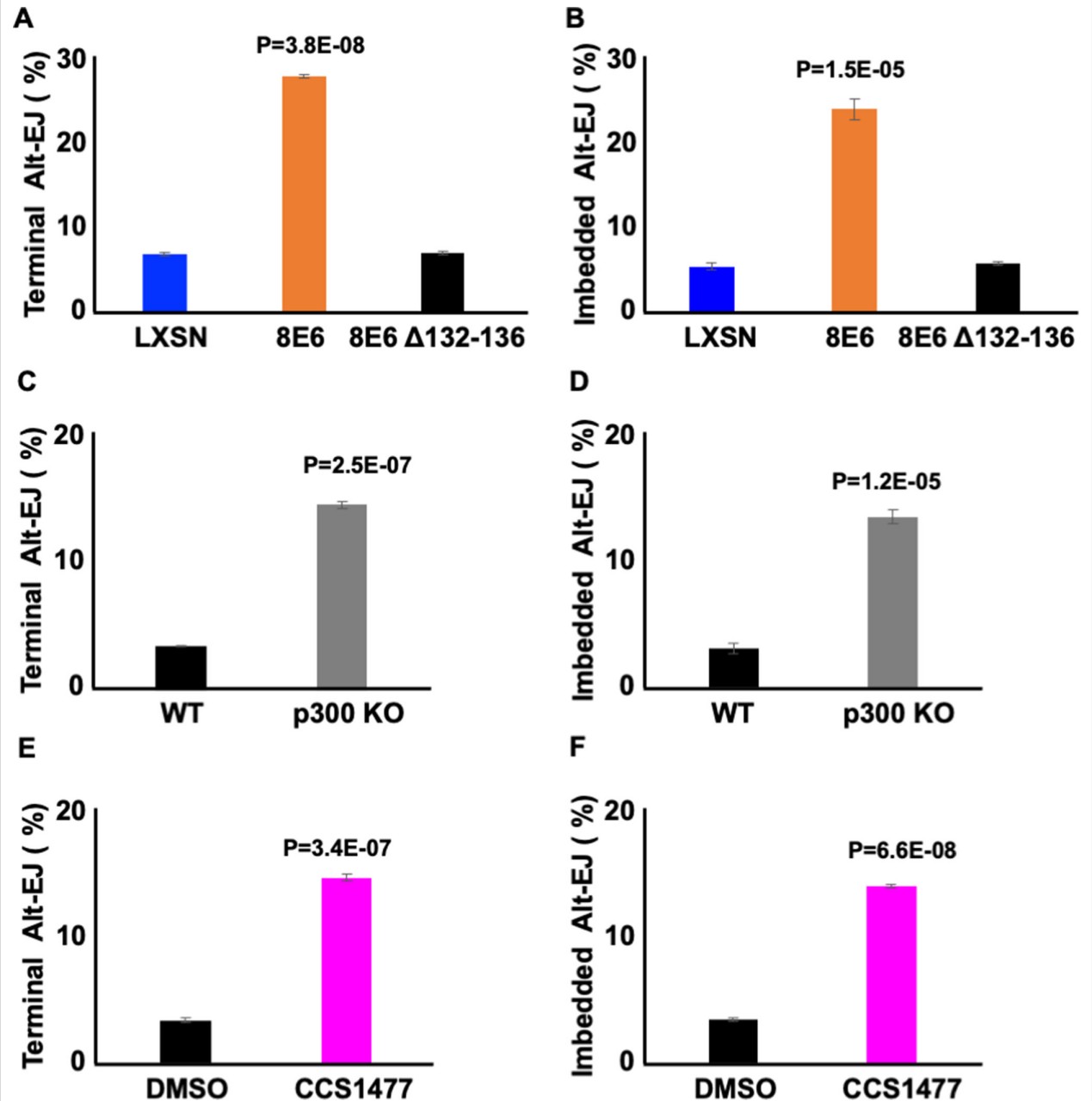

**Figure 8.** Losing p300 activity promotes alternative end joining (Alt-EJ) frequency. (**A–B**) Percentage of U2OS cells that are positive for Alt-EJ following transfection with (**A**) terminal or (**B**) imbedded determined by flow cytometry. (**C–D**) Percentage of human foreskin keratinocyte (HFK) WT and HFK p300 KO cells that are positive for Alt-EJ following transfection with (**C**) terminal or (**D**) imbedded determined by flow cytometry. (**E–F**) Percentage of HFK cells treated with DMSO or CCS1477 (1 μM) that are positive for Alt-EJ following transfection with (**E**) terminal or (**F**) imbedded determined by flow cytometry. All values are represented as mean ± standard error. The statistical significance of differences between cell lines and treatments were determined using Student's t-test. p-Values indicate significant difference between LXSN and 8E6 (**A–B**); WT and p300KO (**C–D**); and DMSO and CCS1477 treatment (**E–F**). Twenty thousand cells were counted for each of three independent flow cytometry experiments.

single-nucleotide polymorphisms (SNPs) and insertions or deletions (indels) were also more for LXSN cells than 8E6 expressing cells with one variation every 3 kb in LXSN compared to one every 3333 bases in 8E6 (*Table 3*).

Repair by Alt-EJ can frequently result in indels with characteristic sequence signatures of flanking microhomology. Using previously established algorithms to identify such characteristic indels (*McIntosh et al., 2020*), we quantified the accumulation of Alt-EJ DNA scars in 8E6 cells compared to LXSN

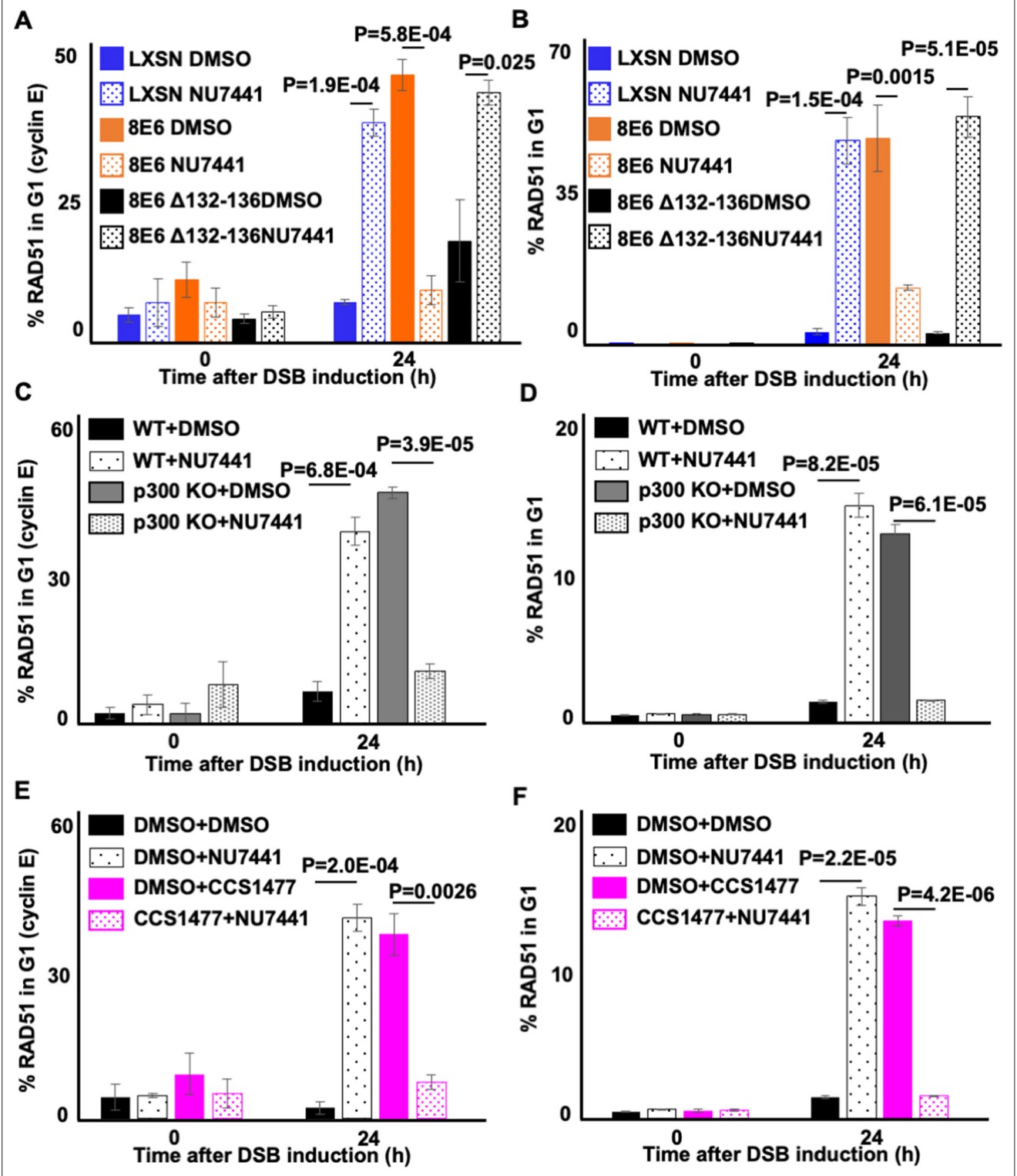

**Figure 9.** NU7441 abrogates RAD51 in G1 induced by losing p300 activity. (**A–B**) Percentage of U2OS treated with DMSO or NU7441 (1 μM) in G1 that RAD51 staining after zeocin treatment (10 μg/mL, 10 min) as determined by (**A**) cyclin E staining or (**B**) flow cytometry. (**C–D**) Percentage of human foreskin keratinocyte (HFK) cells treated with DMSO or NU7441 in G1 that RAD51 staining after zeocin treatment determined by (**C**) cyclin E staining or (**D**) flow cytometry. (**E–F**) Percentage of HFK cells treated with DMSO, CCS1477 (1 μM), or NU7441 in G1 that RAD51 staining after zeocin treatment determined by (**E**) cyclin E staining or (**F**) flow cytometry. All values are represented as mean ± standard error. The statistical significance of differences between treatments were determined using Student's t-test. p-Values indicate significant difference between DMSO and NU7441 with same cell line. At

*Figure 9 continued on next page*

*Figure 9 continued*

least 150 cells were counted over three independent experiments for microscopy. Twenty thousand cells were counted for each of three independent flow cytometry experiments.

control cells. Counting mutations unique to both LXSN and 8E6 cells, this quantified the frequency of short and long deletions with increasing stretches of microhomology up to 20 bp (*Tables 4–6*, *Tables 7 and 8*); small insertions (<5 bp) resulting from templated synthesis in trans previously reported in *Drosophila* (*Tables 4 and 7*, *Table 9*); or large insertions (≥18 bp) (*Tables 4 and 10*) resulting from templated synthesis in cis (snapback synthesis) described in mouse embryonic stem (ES) cells. Deletions represented the most abundant Alt-EJ mutations accumulating in both cell lines. Despite a larger number of annotated indels unique to LXSN (168,005 for LXSN compared to 104,322 for 8E6), this revealed a significantly higher frequency of accumulated short deletions (2–29 bp) in the 8E6 cell line (86.75% in 8E6 compared to 82.33% in LXSN, p-value = 2.2e-16). The frequency of short Alt-EJ type deletions was most significant for deletions with short (≤10 bp) stretches of flanking microhomology (*Table 4*). This trend was similar for long deletions (30–500 bp), albeit not statistically significant (*Table 5*, *Table 6*). Accumulating insertions both small and large bearing Alt-EJ signatures were far less in frequency in both 8E6 and LXSN cell lines though significantly more in LXSN cells (*Tables 7 and 8*) consistent with more non-Alt-EJ mutations annotated in LXSN.

## Discussion

We have previously shown that 8E6 attenuates the two most prominent DSB pathways (HR and NHEJ) (*Hu et al., 2020*; *Wallace et al., 2015*). However, 8E6 delays rather than abrogates DSB repair, leaving the question of how DSBs are repaired in cells expressing 8E6. Here, we show that 8E6 promotes DSB repair via Alt-EJ (*Figure 10*). Because Alt-EJ can be accomplished using different sets of repair factors, we used small molecule inhibitors to determine which Alt-EJ components were most essential for DSB repair in 8E6 expressing cells. This analysis demonstrated that in HFK 8E6 cells, DSBs become three- to fourfold more persistent when PARP1 is inhibited but only ~1.5-fold more persistent when POLθ is inhibited. Thus, 8E6 promotes DSB repair that is dependent on PARP1 and to a lesser extent on POLθ.

Moreover, we show that HFK 8E6 cells can be further induced to repair DSBs via Alt-EJ by the inhibition of an early step during NHEJ (DNA-PKcs) but not a later step in the pathway (Ligase IV). The increased use of Alt-EJ induced by DNA-PKcs inhibition prevented 8E6 from generating previously described DSB repair defects including the formation of RAD51 foci in G1 and delayed DSB repair. We also show that DNA-PKcs inhibition does not cause an increase in HR. Further, we provide mechanistic insight into these phenomena by showing that 8E6 promotes Alt-EJ via p300 degradation and that DNA-PKcs inhibition can prevent RAD51 foci from forming in G1 because of p300 loss. Finally, we provide whole genome sequence analysis that demonstrates 8E6 expression results in a significantly higher frequency of short deletions bearing microhomology signatures of Alt-EJ. These Alt-EJ deletions appear more frequently as short deletions (2–29 bp) bearing short stretches of microhomology (2–10 bp) (*Table 4*).

There are some limitations to this study. This includes potential off target activity of inhibitors. Further complicating the analysis, many DNA repair factors (e.g., PARP1) are involved in more than one repair pathways. We also acknowledge that U2OS may contain mutations (such as an ATR mutation) that change the mechanisms of DSB repair pathway choice (*Elbakry et al., 2021*; *Juhász et al., 2018*). However, the known ways that 8E6 alters DSB repair are consistent in U2OS and primary (HFK) cells, suggesting that results obtained in U2OS cells can be extrapolated to other cell types (*Hu et al., 2020*; *Wallace et al., 2015*). 8E6 was also expressed with the other HPV8 early genes as would be the

**Table 1.** Read counts.

| LXSN | Reads | Quality reads | 8E6 | Reads | Quality reads |
|---|---|---|---|---|---|
| R1 | 480,023,697 | 466,581,986 | R1 | 326,902,764 | 312,236,357 |
| R2 | 480,023,697 | 466,581,986 | R2 | 326,902,764 | 312,236,357 |

**Table 2.** Genome coverage of reads normalized to 624,472,714.

| Samples | Mean read depth | Coverage (%) | Reads mapped (%) |
|---|---|---|---|
| LXSN | 45.4 | 92.4 | 99.74 |
| 8E6 | 44.6 | 92.4 | 99.76 |

case during a natural infection. We are currently working to determine the extent that these other early genes augment or suppress 8E6-induced changes in DSB repair.

Many of the findings in this study are consistent with the existing hypotheses about DSB repair and the potential for beta-HPV infections to promote NMSCs (*Gheit, 2019*; *Kremsdorf et al., 1982*; *Pfister, 2003*). For example, our data in vector control (LXSN) cells show that Alt-EJ is increased when NHEJ is inhibited and that DSB repair is not significantly delayed by inhibition of Alt-EJ. This is compatible with the idea that Alt-EJ is primarily used when there are defects in either HR or NHEJ (*Deriano and Roth, 2013*; *Simsek and Jasin, 2010*; *Truong et al., 2013*). Similarly, by showing that 8E6 promotes Alt-EJ, a mutagenic DSB repair pathway, we provide evidence in support of the idea that transient β-HPV infections may promote tumorigenesis by causing mutations (*Tommasino, 2019*; *Viarisio et al., 2018*; *Wendel and Wallace, 2017*). We also show that p300 restricts DSB repair by Alt-EJ and that at least some of the DSB repair defects caused by p300 loss or 8E6 expression can be overcome by inhibition of DNA-PKcs. Does DNA-PKcs inhibition represent a feasible approach to block the increased mutagenesis associated with 8E6 expression? Or on the contrary, does DNA-PKcs inhibition promote more mutations generated by Alt-EJ?

Induction of Alt-EJ is not limited to HPV8 as a recent report demonstrated that a different HPV protein from a different genus of HPV (HPV16 E7) also increases the use of Alt-EJ (*Leeman et al., 2019*). Further, HPV positive head and neck squamous cell carcinomas and other cancers with down-regulated TGF-b signaling have an elevated frequency of mutations with signatures of repair by Alt-EJ (*Liu et al., 2018*; *Liu et al., 2021*). We show that 8E6 increases the frequency of short deletions with very short stretches of flanking microhomology. Thus, the ability to promote Alt-EJ seems to have evolved in two separate genes in the HPV family (once in HPV16 E7 and once in HPV8 E6) (*Leeman et al., 2019*). Why would these viruses both evolve ways to promote Alt-EJ? Perhaps, this can be linked to their ability to impair HR and/or NHEJ (*Hu et al., 2020*; *Wallace et al., 2015*). Unrepaired DSBs are highly lethal, thus we infer a strong selective pressure for HPV (a non-lytic virus) to find alternative way(s) to repair DSBs.

The data presented here invoke other interesting thoughts. Do HPV8 infections leave Alt-EJ signatures during natural infections? If so, can Alt-EJ signatures be used to provide evidence that naturally occurring beta-HPVs cause mutations? The ability to identify mutations caused by past transient beta-HPV infections would provide the long-sought after evidence that these infections permanently harm host cells.

## Materials and methods

### Key resources table

| Reagent type (species) or resource | Designation | Source or reference | Identifiers | Additional information |
|---|---|---|---|---|
| Cell line (*Homo sapiens*) | HFK | This paper | | Derived from neonatal foreskins |
| Cell line (*Homo sapiens*) | N/TERT HFK | Michael Underbrink (PMID:18256157) | | N/TERT immortalized HFK |
| Cell line (*Homo sapiens*) | U2OS | PMID:10541549 | | Cell line used to measure HR frequency |
| Recombinant DNA reagent | Alt-EJ reporter | Addgene | #113619 | Alt-EJ reporter using 4 nt microhomology |
| Recombinant DNA reagent | Alt-EJ reporter (5′ end) | Addgene | #113620 | sgRNA/CAS9 to induce the 5′ end DSB |
| Recombinant DNA reagent | Alt-EJ reporter (terminal) | Addgene | #113625 #113625 | sgRNA/CAS9 to induce the DSB at the edge of the microhomology |

*Continued on next page*

*Continued*

| Reagent type (species) or resource | Designation | Source or reference | Identifiers | Additional information |
|---|---|---|---|---|
| Recombinant DNA reagent | Alt-EJ reporter (imbedded) | Addgene | #113626 | sgRNA/CAS9 to induce the DSB 8 nt upstream of the microhomology |
| Antibody | Anti-RAD51 (Mouse monoclonal) | Abcam | ab1837 | IF (1:200) |
| Antibody | Anti-cyclin E (Rabbit monoclonal) | Cell Signaling | 4132S | IF (1:200) |
| Antibody | Anti-pH2AX S139 (Rabbit monoclonal) | Cell Signaling | 9718S | IF (1:200) |
| Antibody | Anti-cyclin A (Mouse monoclonal) | Abcam | ab39 | IF (1:200) |
| Antibody | Alexa Fluor 594 (Goat polyclonal) | Thermo Fisher Scientific | A11012 | IF (1:500) |
| Antibody | Alexa Fluor 488 (Goat polyclonal) | Thermo Fisher Scientific | A11001 | IF (1:500) |
| Chemical compound, drug | CCS1477 | Chemietek | CT-CCS1477 | P300 inhibitor |
| Chemical compound, drug | NU7441 | Selleckchem | S2638 | DNA-PKcs inhibitor |
| Chemical compound, drug | Zeocin | Alfa Aesar | J67140-XF | Used to induce DSBs |
| Chemical compound, drug | ART558 | MedChem Express | HY-141520 | Pol Theta inhibitor |
| Chemical compound, drug | DAPI stain | Invitrogen | D1306 | IF (10 µM) |
| Chemical compound, drug | NUCLEAR-ID Red | Enzo Life Science | ENZ-52406 | Flow cytometry (1:1000) |
| Software, algorithm | ImageJ | ImageJ (https://imagej.nih.gov/ij/) | | Version 2.3.0 |
| Software, algorithm | GraphPad Prism | GraphPad Prism (https://graphpad.com) | | Version 9.0.0 |

## Cell culture and reagents

Primary HFKs were derived from neonatal human foreskins. Immortalized human foreskin keratinocytes (N/TERT HFK) provided by Michael Underbrink (University of Texas Medical Branch). Both HFKs were grown in EpiLife medium (MEPICF500, Gibco), supplemented with 60 µM calcium chloride (MEPICF500, Gibco), human keratinocyte growth supplement (MEPICF500, Gibco), and 1% penicillin-streptomycin (PSL02-6X100ML, Caisson). Both keratinocyte cell lines were derived from different donors. They were maintained at low passage number (<20 passages) for all experiments described in this manuscript. Their identity was confirmed by morphology and growth in selective media. Only keratinocytes are viable in EpiLife media.

**Table 3.** Summary of variant analysis reads normalized to 624,472,714.

| Genome (GRCh37) | LXSN | 8E6 |
|---|---|---|
| Number of variants processed | 1,031,891 | 928,763 |
| Number of effects | 2,514,053 | 2,251,061 |
| Genome total length | 3,234,834,690 | 3,234,834,690 |
| Genome effective length | 3,095,677,413 | 3,095,677,413 |
| Variant rate | 1 variant every 3000 bases | 1 variant every 3333 bases |
| Number of annotated genes | 71,845 | 71,078 |
| Insertions | 476,516 | 431,055 |
| Deletions | 555,375 | 497,708 |

**Table 4.** Characteristics of microhomology (Mh) mediated short deletions (2–29 bp) in HFK 8E6 cells.

| LXSN total | Minimum Mh (bp) | Alt-EJ | Frequency (%) | E6 total | Minimum Mh (bp) | Alt-EJ | Frequency (%) | p-Value |
|---|---|---|---|---|---|---|---|---|
| 49,119 | 2 | 40,440 | 82 | 32,763 | 2 | 28,425 | 87 | 2.2e-16 |
| | 3 | 22,580 | 46 | | 3 | 15,512 | 47 | 1.13e-4 |
| | 4 | 16,370 | 33 | | 4 | 11,337 | 35 | 1.61e-4 |
| | 5 | 10,826 | 22 | | 5 | 7664 | 23 | 6.06e-6 |
| | 6 | 8918 | 18 | | 6 | 6468 | 20 | 1.33e-8 |
| | 7 | 7076 | 14 | | 7 | 5037 | 15 | 1.37e-4 |
| | 8 | 6243 | 13 | | 8 | 4557 | 14 | 7.16e-7 |
| | 9 | 4870 | 10 | | 9 | 3507 | 11 | 2.72e-4 |
| | 10 | 4392 | 9 | | 10 | 3165 | 10 | 5.22e-4 |
| | 11 | 3621 | 7 | | 11 | 2498 | 8 | 0.1825 |
| | 12 | 3234 | 7 | | 12 | 2292 | 7 | 0.0222 |
| | 13 | 2448 | 5 | | 13 | 1700 | 5 | 0.1957 |
| | 14 | 2211 | 5 | | 14 | 1522 | 5 | 0.3412 |
| | 15 | 1847 | 4 | | 15 | 1185 | 4 | 0.2958 |
| | 16 | 1602 | 3 | | 16 | 1060 | 3 | 0.8522 |
| | 17 | 1238 | 3 | | 17 | 763 | 2 | 0.0861 |
| | 18 | 1083 | 2 | | 18 | 682 | 2 | 0.2440 |
| | 19 | 866 | 2 | | 19 | 509 | 2 | 0.0240 |
| | 20 | 765 | 2 | | 20 | 467 | 1 | 0.1358 |

U2OS were maintained in DMEM supplemented with 10% FBS and 1% penicillin-streptomycin. Zeocin (J67140-XF, Alfa Aesar) was used to induce DSBs (10 µg/mL, 10 min). Their identity was confirmed by the presence of the DR-GFP construct used to measure HR. This reporter cassette only exists in these cells within our lab. 8E6 expression (or lack thereof) was confirmed by rtPCR. All cell lines in this study underwent regular mycoplasma testing. NU7441 (S2638, Selleckchem) was used to inhibit DNA-PKcs phosphorylation (1 µM) and verify the pDNA-PKcs antibody. siRNA DNA-PKcs was used to further validate pDNA-PKcs antibody. KU55933 (Sigma-Aldrich, SML1109) was used to validate RAD51 antibody as previously described (*Bakr et al., 2015*). CCS1477 (CT-CCS1477, Chemietek) was used to inhibit p300 activity (1 µM). ART558 (HY-141520, MedChem Express) was used to inhibit Pol Theta activity. Alt-EJ plasmids (#113619, #113620, #113625, #113626, Addgene) were used to measure Alt-EJ efficiency.

## Immunofluorescence microscopy

Cells were seeded onto either 96-well glass-bottom plates and grown overnight. Cells treated with zeocin (10 µg/mL, 10 min) were fixed with 4% paraformaldehyde. Then, 0.1% Triton-X was used to permeabilize the cells, followed by blocking with 3% bovine serum albumin. Cells were then incubated with the following antibodies: RAD51 (ab1837, Abcam, 1:200), cyclin E (4132S, Cell Signaling), pH2AX S139 (9718S, Cell Signaling), and cyclin A (ab39, Abcam). The cells were washed and stained with the appropriate secondary antibodies: Alexa Fluor 594 (red) goat anti-rabbit (A11012, Thermo Fisher Scientific), Alexa Fluor 488 (green) goat anti-mouse (A11001, Thermo Fisher Scientific). After washing, the cells were stained with 10 µM DAPI in PBS and visualized with the Zeiss LSM 770 microscope. Images were analyzed using the ImageJ techniques previously described (*Murthy et al., 2018*). Cyclin E intensity was measured for each cell. Average cyclin E intensity of cells grown in media without growth factor for 4 hr was used to define the threshold of cyclin E positive.

**Table 5.** Characteristics of microhomology (Mh) long deletions (30–500 bp) in human foreskin keratinocyte (HFK) 8E6 cells.

| LXSN total | Minimum Mh (bp) | Alt-EJ | Frequency (%) | E6 total | Minimum Mh (bp) | Alt-EJ | Frequency (%) | p-Value |
|---|---|---|---|---|---|---|---|---|
| 2612 | 2 | 1959 | 75 | 1168 | 2 | 888 | 76 | 0.5247 |
| | 3 | 1844 | 71 | | 3 | 842 | 72 | 0.3704 |
| | 4 | 1735 | 66 | | 4 | 782 | 67 | 0.7790 |
| | 5 | 1689 | 65 | | 5 | 765 | 65 | 0.6460 |
| | 6 | 1635 | 63 | | 6 | 722 | 62 | 0.6734 |
| | 7 | 1594 | 61 | | 7 | 693 | 59 | 0.3429 |
| | 8 | 1536 | 59 | | 8 | 673 | 58 | 0.5171 |
| | 9 | 1487 | 57 | | 9 | 656 | 56 | 0.6868 |
| | 10 | 1433 | 55 | | 10 | 634 | 54 | 0.7669 |
| | 11 | 1381 | 53 | | 11 | 616 | 53 | 0.9684 |
| | 12 | 1332 | 51 | | 12 | 596 | 51 | 1.0000 |
| | 13 | 1286 | 49 | | 13 | 568 | 49 | 0.7580 |
| | 14 | 1245 | 48 | | 14 | 543 | 46 | 0.5265 |
| | 15 | 1202 | 46 | | 15 | 530 | 45 | 0.7410 |
| | 16 | 1168 | 45 | | 16 | 505 | 43 | 0.4172 |
| | 17 | 1131 | 43 | | 17 | 485 | 42 | 0.3249 |
| | 18 | 1089 | 42 | | 18 | 468 | 40 | 0.3673 |
| | 19 | 1051 | 40 | | 19 | 451 | 39 | 0.3644 |
| | 20 | 1010 | 39 | | 20 | 433 | 37 | 0.3697 |

## Flow cytometry

Cells were collected from 6 cm plates, at about 80–90% confluence, by using trypsinization. Cells were washed with cold PBS and fixed with 95% cold ethanol for 10 min at –20°C. Cells were stained with anti-RAD51 antibody (ab1837, Abcam, 1:100) and Alexa Fluor 488 goat anti-mouse (A11001, Thermo Fisher Scientific). After washing, cells were resuspended in 200 µL PBS and NUCLEAR-ID Red DNA stain (ENZ-52406, Enzo Life Science), and incubated in the dark at room temperature for 30 min. Samples were analyzed by a BD Accuri C6 Plus Flow Cytometer.

## Immunoblotting

After being washed with ice-cold PBS, cells were lysed with RIPA Lysis Buffer (VWRVN653-100ML, VWR Life Science), supplemented with Phosphatase Inhibitor Cocktail 2 (P5726-1ML, Sigma) and Protease Inhibitor Cocktail (B14001, Bimake). The Pierce BCA Protein Assay Kit (89167-794, Thermo Fisher Scientific) was used to determine protein concentration. Equal protein lysates were run on Novex 3–8% Tris-acetate 15 Well Mini Gels (EA03785BOX, Invitrogen) and transferred to Immobilon-P membranes (IPVH00010, Thermo Fisher Scientific). Membranes were then probed with the following primary antibodies: GAPDH (sc-47724,

**Table 6.** Filtered variants for Alt-EJ analyses.

| | LXSN | 8E6 |
|---|---|---|
| Pre-filtered | 879,302 | 820,766 |
| Unique INDELs | 168,005 | 104,322 |

**Table 7.** Short deletions bearing microhomology signatures of Alt-EJ.

| Sample | Short deletions (2–29 bp) | Matching Alt-EJ | Frequency (%) | p-Value |
|---|---|---|---|---|
| LXSN | 49,119 | 40,440 | 82.33 | |
| 8E6 | 32,763 | 28,425 | 86.75 | 2.2e-16 |

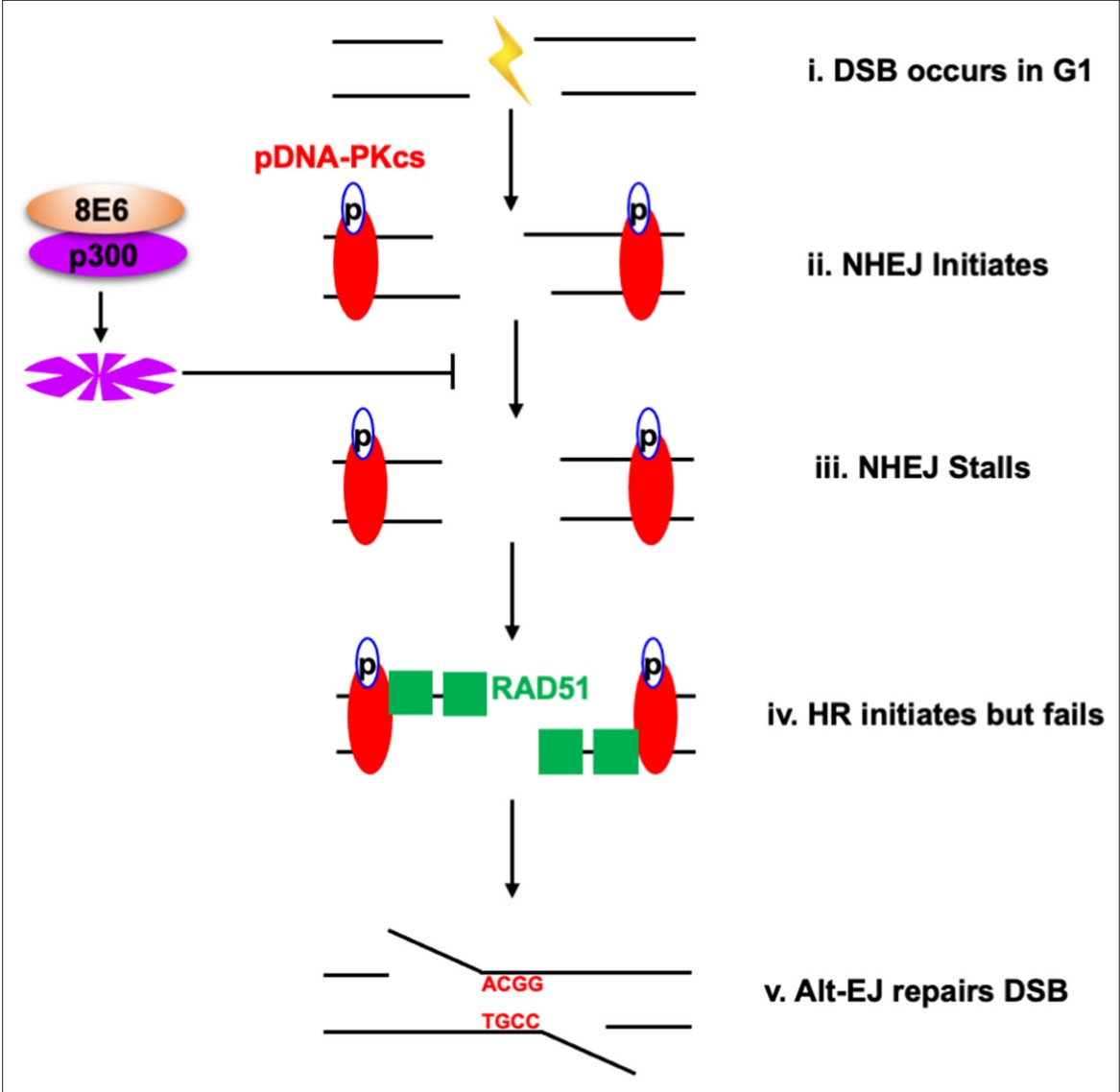

**Figure 10.** Alternative end joining (Alt-EJ) repairs double strand breaks (DSBs) in cells expressing 8E6. (**i**) DSB occurs in G1 phase in cells expressing 8E6. (**ii**) Non-homologous end joining (NHEJ) initiates with auto-phosphorated DNA-dependent protein kinase catalytic subunit (DNA-PKcs). (**iii**) 8E6 stalls NHEJ by degrading p300 (*Hu et al., 2020*). (**iv**) Homologous recombination (HR) initiates and fails at the site of failed NHEJ (*Hu and Wallace, 2022c*). (**v**) Finally, PARP-1-dependent Alt-EJ repairs the DSB, which lead to microhomology mediated indels.

Santa Cruz Biotechnologies, 1:1000) P300 (sc-48343, Santa Cruz Biotechnologies). CAS9 (65832S, Cell Signaling). Pol Theta (PA5-69577, Thermo Fisher Scientific). After exposure to the matching HRP-conjugated secondary antibody, cells were visualized using SuperSignal West Femto Maximum Sensitivity Substrate (34095, Thermo Fisher Scientific).

## Transfection and Alt-EJ assay

HFK cells were plated in 3 mL of complete growth medium in a 6 cm plate. Cells were used at 80% confluency. Two µg of plasmids were diluted in 200 µL Xfect transfection reagent (631317, Takara). The mixture was incubated at room temperature for 15 min. The transfection mixture was added to each well drop-wise and incubated for 48 hr at 37°C. Cells were harvested for flow cytometry analysis.

## Whole genome sequencing analysis

HFK cells were transfected with the 8E6 expression vector or empty vector and equally passaged under selective pressure to establish HFK 8E6 versus control HFK LXSN cells. DNA was extracted

**Table 8.** Long deletions bearing microhomology signatures of Alt-EJ.

| Sample | Long deletions (≥30 bp) | Matching Alt-EJ | Frequency (%) | p-Value |
|---|---|---|---|---|
| LXSN | 2612 | 1959 | 75.00 | |
| 8E6 | 1168 | 888 | 76.03 | 0.5247 |

**Table 9.** Short insertions bearing microhomology signatures of Alt-EJ.

| Sample | Short insertions (<5 bp) | Matching Alt-EJ | Frequency (%) | p-Value |
|---|---|---|---|---|
| LXSN | 50,390 | 959 | 1.90 | 5.62e-10 |
| 8E6 | 27,611 | 367 | 1.33 | |

using Trizol (Invitrogen), libraries were prepared using the Illumina DNA PCR-Free prep kit and sequenced with paired 300-bp v1.5 reads on a Novaseq 6000. All protocols were performed using the standard procedures provided by the manufacturers. Sequences have been deposited in the NCBI SRA database with accession number (PRJNA 856469).

## Variant calling and Alt-EJ mutational analysis

Whole genome sequences for passage matched LXSN and 8E6 cell lines were analyzed for indel detection using the best practices for variant discovery analysis outlined by the Broad Institute (*DePristo et al., 2011*). Briefly, reads from the sequenced samples were preprocessed for quality control by removing adapter sequences and low-quality reads (<Q20). Quality paired-end reads from LXSN were normalized to the same number of reads from 8E6 (624,472,714) and both sets were mapped to the human reference genome (GRCh37/hg19) using Burrows-Wheeler Aligner (BWA v0.7.17) (*Li and Durbin, 2010*). Aligned reads were sorted and duplicate reads were removed using Picard-tools (v2.25.5). Raw genomic variants, including SNPs and DNA insertions and deletions (indels), were identified using the GATK HaplotypeCaller algorithm in GATK (v4.1.9.0) ('Genomics in the Cloud [Book]', n.d.). The identified indels were extracted and filtered for quality with GATK VariantFiltration (QUAL score normalized by allele depth: QD <2.0, Phred-scaled probability: FS >200, and Symmetric odds ratio test: SOR >10), followed by base quality score recalibration using GATK BaseRecalibrator and applied to the aligned bam files. A second round of variant calling (GATK Haplotype Caller) was performed using recalibrated (analysis-ready) SNPs and indels further filtered for quality. SnpEff 5.0 was used to annotate the genetic variants.

Alt-EJ mutational signatures were counted as previously described (*McIntosh et al., 2020*). Briefly, indels previously identified by the 1000 Genomes Project or shared between LXSN and 8E6 cell lines were removed. Alt-EJ mutational signatures bearing flanking microhomology were then determined in R for small deletions (2–29 bp) and long deletions (30–500 bp), small insertions (<5 bp), and large snap-back insertions (≥18 bp). For analysis of deletions with microhomology, the sequence adjacent to each deletion was extracted based on the human genome (R package BSgenome.Hsapiens. UCSC. hg19) and annotated according to the number of contiguous matching nucleotides between the deletion and the adjacent sequence, counted from the beginning of each sequence (microhomology). Different microhomology length thresholds (ranging from 2 to 20 bp) were then used to filter the results into Alt-EJ matching deletions unique to each LXSN and E6 cell line.

To detect synthesis-dependent small insertions, we extracted ±15 nt surrounding the insertion (from the hg19 human genome) and identified whether there was a repeated sequence either 5′ or 3′ to the insertion that matched the full insertion and also at least 2 nt on either side. Matches also needed to have a gap of at least 1 nt between the two sequences.

For synthesis-dependent snapback (large) insertions, only the insertion sequence was considered. Each sequence was compared to its own reverse-complement sequence and any repeats of ≥7 nt also separated by at least 4 nt were labeled as snapback-like. To avoid spurious false negatives, only insertions of at least (2×7)+4 = 18 nt were considered.

## Cell viability assay

Ten thousand cells/well were seeded on a 96-well plate and grown for 24 hr. Zeocin treatment (10 μg/mL, 10 min) was applied 24 hr after

**Table 10.** Long insertions bearing microhomology signatures of Alt-EJ.

| Sample | Long insertions (≥18 bp) | Matching Alt-EJ | Frequency % | p-Value |
|---|---|---|---|---|
| LXSN | 5485 | 1068 | 19.47 | 1.547e-05 |
| 8E6 | 2545 | 392 | 15.40 | |

seeding. Then ART558 dose series was added and incubated for 48 hr at 37°C. Forty-eight hr after treatment, 10 µL/well of MTT solution (10 mg/mL) was added for 24 hr. Subsequently, wells were incubated with 100 µL solubilization solution for 24 hr and the optical density measured at 640 nm. Doses for ARTT558 : 0, 1, 5, 10, 15, 20, 25, 50.

## Statistical analysis

All values are represented as mean ± standard error (SE). Statistical differences between groups were measured by using Student's t-test. p-Values in all experiments were considered significant at less than 0.05. Statistical significance of Alt-EJ mutational frequencies between LSXN and 8E6 in each instance (small or large deletions or insertions) were determined by Fisher's exact t-test using prop.test in R.

## Acknowledgements

We appreciate KSU-CVM Confocal Core for our immunofluorescence microscopy, Michael Underbrink for providing the TERT-immortalized HFKs, and Jeremy M Stark for providing plasmids of the alt-EJ reporter assay. This research was supported by the National Institute of General Medical Sciences (NIGMS) of the National Institutes of Health (NIH, P20GM130448); NIH Research Enhancement Award (NCI R15 CA242057 01A1); US Department of Defense (CMDRP PRCRP CA160224 [NW]); and Johnson Cancer Research Center of Kansas State University. We acknowledge the University of Kansas Medical Center Genomics Core for your core support services in publications containing Genomics Core generated data. We would also like to note the following NIH supporting grants – Kansas Intellectual and Developmental Disabilities Research Center (NIH U54 HD 090216), the Molecular Regulation of Cell Development and Differentiation – COBRE (P30 GM122731-03) – the NIH S10 High-End Instrumentation Grant (NIH S10OD021743) and the Frontiers CTSA grant (UL1TR002366) at the University of Kansas Medical Center, Kansas City, KS 66160. We would also like to acknowledge Dr Massimo Tommasino. His kindness and influence will be missed.

## Additional information

### Funding

| Funder | Grant reference number | Author |
| --- | --- | --- |
| National Institute of General Medical Sciences | P20GM130448 | Nicholas Wallace |
| National Institutes of Health | NCI R15 CA242057 01A1 | Nicholas Wallace |
| U.S. Department of Defense | CMDRP PRCRP CA160224 (NW) | Nicholas Wallace |

The funders had no role in study design, data collection and interpretation, or the decision to submit the work for publication.

### Author contributions

Changkun Hu, Conceptualization, Data curation, Software, Formal analysis, Investigation, Visualization, Methodology, Writing - original draft; Taylor Bugbee, Formal analysis, Validation, Investigation, Visualization, Writing - review and editing; Rachel Palinski, Software, Formal analysis, Investigation, Methodology, Writing - review and editing; Ibukun A Akinyemi, Formal analysis; Michael T McIntosh, Sumita Bhaduri-McIntosh, Formal analysis, Writing - review and editing; Thomas MacCarthy, Methodology; Nicholas Wallace, Conceptualization, Resources, Supervision, Funding acquisition, Investigation, Writing - review and editing

### Author ORCIDs

Changkun Hu http://orcid.org/0000-0002-4407-7144
Nicholas Wallace http://orcid.org/0000-0002-3971-716X

**Decision letter and Author response**
Decision letter https://doi.org/10.7554/eLife.81923.sa1
Author response https://doi.org/10.7554/eLife.81923.sa2

## Additional files

### Supplementary files
• MDAR checklist

### Data availability
Sequences have been deposited in the NCBI SRA database with accession number (PRJNA 856469).

The following dataset was generated:

| Author(s) | Year | Dataset title | Dataset URL | Database and Identifier |
|---|---|---|---|---|
| Palinski R | 2022 | Beta human papillomavirus 8E6 promotes alternative end-joining | https://www.ncbi.nlm.nih.gov/bioproject/?term=PRJNA856469 | NCBI BioProject, PRJNA856469 |

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
