## [Editor Report]

This article reports useful data on how human papillomavirus 8E6 protein regulates DSB repair pathways in human cells. The data support the claim that 8E6 promotes alternative end-joining through binding and destabilizing the p300 acetyltransferase, showing the involvement of PARP-1-dependent alternative end-joining and to a lesser degree DNA Polymerase theta-dependent alternative end-joining.

---

## [Decision Letter]

**Decision letter after peer review:**

Thank you for submitting your article "Β human papillomavirus 8E6 promotes alternative end-joining" for consideration by *eLife*. Your article has been reviewed by 3 peer reviewers, including Wolf-Dietrich Heyer as the Reviewing Editor and Reviewer #1, and the evaluation has been overseen by Jessica Tyler as the Senior Editor.

Recommendations for the authors:

This study reports convincing data that in 8E6 expressing cells DSB repair is shifted towards alt-EJ at the expense of NHEJ and HR, following logically previous published studies with HPV16 and earlier work that 8E6 inhibits NHEJ and HR but does not preempt DSB repair. It is important to show experimentally, what might be considered evident, but the advance remains rather incremental. In addition, the study remains descriptive and offers little insight into the mechanisms involved other than it involves and binding (and inferred destruction) of p300. How p300 regulates DSB repair pathway choice remains unclear and is not further illuminated by this study. To add some novelty that goes beyond the HPV16 studies and closing an evident gap in the argument it to demonstrate an involvement of DNA polymerase theta to ascertain that the alt-EJ pathway involved is TMEJ. While most essential revisions require only text changes or clarifications, #4 is critical and will involve novel experimentation testing the involvement of DNA polymerase theta.

Essential revisions:

1) Abstract, line 24: There appear to be several alt-EJ pathways. Does this refer to TMEJ? If so, it seems now to be clear that TMEJ has a unique function and is not a backup pathway as evinced by the POLq KO phenotype and recent publications on POLq action in mitosis. See also lines 90 and 108.

2) Please address the following points either by text changes or a rebuttal:

Introduction:

Leeman et al. 2019 and Liu et al. 2018 and 2021 are precedents and should be in the introduction.

Lines 263-264:

The sentence is self-serving as Leeman et al. 2019 and Liu et al. 2018 are precedents.

Line 266:

Weaver et al., 2015 is incorrectly referenced. The paper reports that HPV+ HNSCC are sensitive to PARP inhibition but does not report a mutational signature or evidence of alt-EJ.

Lines 265-66:

The correlation of a mutational signature consistent with increased Alt-EJ across TCGA was reported by Liu et al. 2021.

Lines 267:

The statement that 8E6 causes a similar signature is a tautology.

Lines 269-73:

The statement that the ability to promote Alt-EJ seems to have evolved at least twice independently in the HPV family needs more clarification-Isn't E6 similar in both HPV16 and HPV8? And given that it is loss of HR/NHEJ that promotes the use of alt-EJ, the statement (line 273) that there is "strong selective pressure for HPV (a non-lytic virus) to find a way to repair DSBs" is not correct.

3) Primary and h-TERT immortalized human foreskin keratinocytes are not sufficiently described-is this one or multiple sources, at what passage were experiments conducted, was mycoplasma testing routine, how was cell identity verified? Similar questions pertain to U20S.

4) It seems critical to experimentally test an involvement of DNA polymerase theta (POLq) in the 8E6-induced alt-EJ events reported here. This does not require showing POLq dependence for each and every assay but available POLq KO cell lines or knockdown systems should test this point for some of the endpoints reported here.

This will also require an additional paragraph in the introduction and discussion to highlight the complexity of alt-EJ pathways to put TMEJ in context.

5) Can the authors explain the quite large differences in Alt-EJ activity between figures; e.g., in figure 1 the imbedded activity is 12.46% versus 22% in figure 2. Is this transfection efficiency? A note in the manuscript may be required to explain this.

6) The alt-EJ assay (Bhargava et al.) should be discussed in more detail and the relevant figure should show the nucleotide sequences involved. In particular, the involvement of the 4 nt microhomology should be discussed, as for this event the assay is significantly POLq-dependent (Figure 4 of Bhargava et al.). This becomes also important in the discussion of the genomic DNA sequencing data.

7) In a publication too recent for the author to include but relevant for the interpretation and discussion of their PARPi data, Luedeman et al. (2022 Nature Comm PMID: 35927262) showed a rather modest effect of PARP on TMEJ. In a revision, this publication should be included and discussed.

8) Figure 4AB. The results are not well described in the text. Isn't the key result that lesions (γ H2AX foci) are down in 8E6 expressing cells?

[Editors' note: further revisions were suggested prior to acceptance, as described below.]

Thank you for resubmitting your work entitled "Β human papillomavirus 8E6 promotes alternative end-joining" for further consideration by *eLife*. Your revised article has been evaluated by Paivi Ojala (Senior Editor) and a Reviewing Editor.

The manuscript has been improved but there are some remaining issues that need to be addressed, as outlined below:

Recommendations for the authors:

The authors addressed the concerns of the reviewers by adding the requested experiments testing the involvement of DNA polymerase theta and making the necessary text changes and adding clarifications.

The following changes are required before acceptance:

Figure 2 supplement 1A and C: Add some white space between the 2 blots to indicate that these are separate.

Please clarify and make sure that the loading controls (GAPDH) were run on the same gel from which the Cas9 signal was derived in Figure 2 supplement 1A and C.

---

## [Author Response]

Essential revisions:1) Abstract, line 24: There appear to be several alt-EJ pathways. Does this refer to TMEJ? If so, it seems now to be clear that TMEJ has a unique function and is not a backup pathway as evinced by the POLq KO phenotype and recent publications on POLq action in mitosis. See also lines 90 and 108.

We thank the reviewers for this suggestion. The experiments suggested in essential revision 4 allowed us to determine that at least some of the DSB repair promoted by 8E6 requires POLθ. However, 8E6 did not increase POLθ abundance. We have now changed the language in the abstract and have added a figure examining the dependence of DSB repair on POLθ activity in 8E6 expressing cells. Because these data did not show that all of the DSB repair in 8E6 expressing cells occurred in a POLθ-dependent manner, we have chosen not to switch to the term TMEJ. We do not know that if our use of the term Alt-EJ includes all the mechanisms detectable by the reporter assays, including TMEJ. Please see abstract and discussion of the new figure 3.

2) Please address the following points either by text changes or a rebuttal:Introduction:Leeman et al. 2019 and Liu et al. 2018 and 2021 are precedents and should be in the introduction.

These references are now mentioned in the discussion to provide context to our discovery. We are open to moving them to the introduction if desired but preferred this approach for organizational reasons. See line 303-308.

Lines 263-264:The sentence is self-serving as Leeman et al. 2019 and Liu et al. 2018 are precedents.

This line has been removed. See line 303-304.

Line 266:Weaver et al., 2015 is incorrectly referenced. The paper reports that HPV+ HNSCC are sensitive to PARP inhibition but does not report a mutational signature or evidence of alt-EJ.

We have removed this reference.

Lines 265-66:The correlation of a mutational signature consistent with increased Alt-EJ across TCGA was reported by Liu et al. 2021.

We now mention this fact. See line 303-308.

Lines 267:The statement that 8E6 causes a similar signature is a tautology.

This has been fixed. See line 303-308.

Lines 269-73:The statement that the ability to promote Alt-EJ seems to have evolved at least twice independently in the HPV family needs more clarification-Isn't E6 similar in both HPV16 and HPV8? And given that it is loss of HR/NHEJ that promotes the use of alt-EJ, the statement (line 273) that there is "strong selective pressure for HPV (a non-lytic virus) to find a way to repair DSBs" is not correct.

We respectfully disagree with this conclusion. There are similarities between the E6s from HPV8 and HPV 16. However, Leeman et al. demonstrated that HPV16 E7 promotes the use of Alt-EJ, while we show that HPV8 E6 promotes the pathway. Thus, in a member of one genus of HPV (β-papillomaviruses) the E6 protein evolved to promote Alt-EJ, while in another genus of HPV (α-papillomaviruses) the E7 protein evolved to promote Alt-EJ. We have modified our language to make this point clearer and to soften our claims as we acknowledge that we have no evolutionary evidence. Please see line 303-309.

3) Primary and h-TERT immortalized human foreskin keratinocytes are not sufficiently described-is this one or multiple sources, at what passage were experiments conducted, was mycoplasma testing routine, how was cell identity verified? Similar questions pertain to U20S.

Primary and hTERT-immortalized human foreskin keratinocytes (HFK) were derived from separate donors. All cell lines in this study underwent regular mycoplasma testing. HFK cell line identity was confirmed by growth in restrictive growth media. The identity of U2OS cells was confirmed based on the presence of the integrated DR-GFP cassette used to measure homologous recombination. They are the only cell line in our lab that contains this cassette. This information along with approximate passage numbers for HFK is now included in the resubmission as part of the methods section. We are unable to provide passage information for the U2OS cells as this information was not provided to the π when he acquired them as a post-doctoral fellow. Please see lines 322-335 and Key Resources Table.

4) It seems critical to experimentally test an involvement of DNA polymerase theta (POLq) in the 8E6-induced alt-EJ events reported here. This does not require showing POLq dependence for each and every assay but available POLq KO cell lines or knockdown systems should test this point for some of the endpoints reported here.This will also require an additional paragraph in the introduction and discussion to highlight the complexity of alt-EJ pathways to put TMEJ in context.

We appreciate this suggestion from the reviewers and have taken a three-pronged approach to address it: (i) performing western blot analysis of Pol Theta to determine if 8E6 changed the abundance of the protein, (ii) determining if Pol Theta inhibition by small molecule specifically impeded DSB repair (microscopy of pH2AX) in 8E6 expressing HFKs, and (iii) determining if 8E6 made cells dependent on Pol Theta for survival. These data demonstrated that 8E6 promotes the use of TMEJ, but that it does not cause cells to become exclusively reliant on the pathway. The description of these results and the data itself can now be found in Figure 3 and lines 115-139. It is also discussed in lines 262-267. We should note that instead of using the cell lines suggested that we used HFKs and an inhibitor because they are more biologically relevant.

5) Can the authors explain the quite large differences in Alt-EJ activity between figures; e.g., in figure 1 the imbedded activity is 12.46% versus 22% in figure 2. Is this transfection efficiency? A note in the manuscript may be required to explain this.

The reviewers are correct. The differences in magnitude are the result of differences in transfection efficiency. This motivated us to correct for transfection efficiency during our initial analysis (Figure 1—figure supplement 1, Figure 2—figure supplement 1, and Figure 4—figure supplement 1). Unfortunately, we did not mention this in our original submission. We now note how we accounted for differences in transfection efficiency. Please see line 100-102.

6) The alt-EJ assay (Bhargava et al.) should be discussed in more detail and the relevant figure should show the nucleotide sequences involved. In particular, the involvement of the 4 nt microhomology should be discussed, as for this event the assay is significantly POLq-dependent (Figure 4 of Bhargava et al.). This becomes also important in the discussion of the genomic DNA sequencing data.

We added the requested details for the Alt-EJ assay. Please see Figure1 legend and line 98.

7) In a publication too recent for the author to include but relevant for the interpretation and discussion of their PARPi data, Luedeman et al. (2022 Nature Comm PMID: 35927262) showed a rather modest effect of PARP on TMEJ. In a revision, this publication should be included and discussed.

We have added a discussion of the data described in Luedeman et al. and thank the reviewers for bringing this to our attention. Please see lines 116-118.

8) Figure 4AB. The results are not well described in the text. Isn't the key result that lesions (γ H2AX foci) are down in 8E6 expressing cells?

We have made the requested changes. Please see lines 159-162.

[Editors' note: further revisions were suggested prior to acceptance, as described below.]

The manuscript has been improved but there are some remaining issues that need to be addressed, as outlined below:Recommendations for the authors:The authors addressed the concerns of the reviewers by adding the requested experiments testing the involvement of DNA polymerase theta and making the necessary text changes and adding clarifications.The following changes are required before acceptance:Figure 2 supplement 1A and C: Add some white space between the 2 blots to indicate that these are separate.

Thank you for this suggestion. We added space between blots for all figures. We also added outlines for blots to make it clear.

Please clarify and make sure that the loading controls (GAPDH) were run on the same gel from which the Cas9 signal was derived in Figure 2 supplement 1A and C.

We confirm that all loading controls (GAPDH) were run on the same gel from which the CAS9 signals were derived. This is correct for all figures.